# Unmasking the Success of Mixture of Experts: Noisy Features, Sparse Experts

## Abstract

Mixture of Experts (MoEs) allow deep neural networks to grow in size without incurring large inference costs. Theories to explain their success largely focus on enhanced expressiveness and generalization compared to single expert models. We identify a novel mechanism: MoEs can outperform single experts by utilizing activation sparsity amidst feature noise, even without increase in parameter count. This enables MoEs to achieve superior generalization performance, robustness, training convergence speed and sample complexity. Our results further offer a theoretical basis for techniques like MoEfication, which transform dense layers into MoE architectures by exploiting activation sparsity. Experiments on synthetic data and standard real-world language tasks support our theoretical insights.

## 1 Introduction

Transformer-based models have achieved significant success in a range of deep learning tasks, including natural language processing (Brown et al., 2020; Achiam et al., 2023; Yang et al., 2024) and computer vision (Dosovitskiy et al., 2020; Liu et al., 2021). Since the performance of these models often scales with the number of parameters (Kaplan et al., 2020), the field has seen the development of ever larger models, such as GPT-4 with over a trillion parameters (Achiam et al., 2023). However, this rapid growth has escalated computational demands, training costs, and environmental impact (Strubell et al., 2020). In response, Mixture-of-Experts (MoE) models have emerged as a class of neural networks designed for efficient scalability (Jacobs et al., 1991; Shazeer et al., 2017). Unlike traditional "dense" models that activate all their parameters for every input, MoE architectures are composed of smaller sub-networks, called "experts", and utilize conditional computation (Bengio et al., 2015). For any given input, a gating network (also called router) selects only a fraction of these experts to be evaluated and aggregated. This approach, exemplified by models like DeepSeek-V3 (Liu et al., 2024a) and Mixtral 8×7B (Jiang et al., 2024), allows MoE models to outperform their dense counterparts with the same computational resources during inference because computation scales with the size of the activated experts, not the total model size (Fedus et al., 2022).

Despite their efficiency at inference time, training large-scale MoE models from scratch presents significant challenges, including substantial memory requirements and training instability. To address these difficulties, a growing area of research focuses on converting pre-trained dense models into MoE architectures—a process sometimes referred to as "MoEfication" (Zhang et al., 2021), which has led to models like MoEBERT (Zuo et al., 2022) and LLaMA-MoE (Zhu et al., 2024). This conversion allows the resulting MoE model to retain the same parameter count as the original dense model, enabling a direct speedup in inference due to the inherent structural advantages of the MoE design. Historically, this process capitalized on the natural activation sparsity of ReLU-based models (Li et al., 2022b). However, modern large language models predominantly use soft activation functions (e.g., SwiGLU (Shazeer, 2020), GeGLU (Team et al., 2024)) that exhibit significantly lower natural sparsity. Although sparsity can be re-introduced through methods like continued pre-training with ReLU (Mirzadeh et al., 2023; Song et al., 2024a), this can be computationally expensive. Our investigation is motivated by the observation that even within dense activation patterns, a latent modular structure still exists. As illustrated in Figure 1, this underlying structure can be revealed through techniques like activation pruning.

The main objective of this paper is to theoretically explain how such activation sparsity can be exploited by MoE architectures, in particular, in the presence of feature noise. As we show, it does

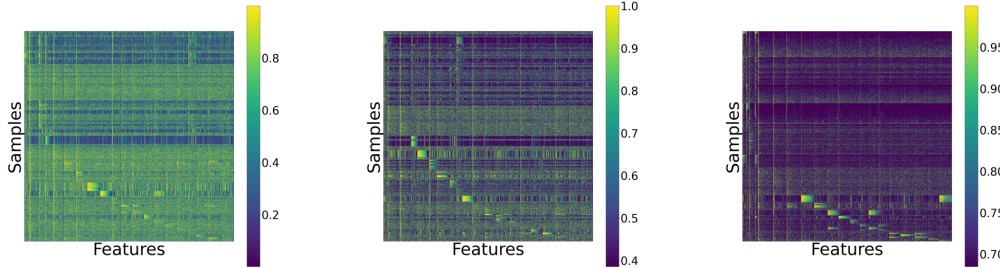

Figure 1: Modular structure in input activations to the up_proj layer within the MLP block of *layer-0* of the Llama-2-7B model, revealed using TEAL (Liu et al., 2024b), for uniform magnitude pruning. From left to right, panels show activation percentiles after pruning activations based on 0%, 40%, and 70% *uniform* activation sparsity thresholds, respectively. Since the activation distributions are skewed due to the presence of outlier activations, we plot activation percentiles.

not only speed up inference, it also leads to improved generalization, better sample complexity, and enhanced robustness. As a consequence, we identify a novel mechanism how MoEs can outperform dense counterparts. This mechanism is rooted in the introduction of feature noise within the activation values of the neural network. This stands in contrast to existing MoE theory that primarily focuses on gains from increased parameter counts or assumes a clustered data structure that is more suitable for a MoE representation (Chen et al., 2022). Our theoretical analysis establishes that even when an MoE and its dense counterpart have the same number of parameters and are equally capable of representing the ground truth function, the MoE's structure provides a distinct advantage in noisy environments. In doing so, we also provide a theoretical justification why MoEfication could can be advantageous beyond improved inference speed.

**Contributions** In summary, we make the following contributions:

- We identify a novel mechanism how MoEs can outperform their dense counterpart: They can be more robust to feature noise.
- In contrast to contemporary theory on MoEs, we investigate not only generalization performance, but also highlight other benefits: enhanced robustness to certain input perturbations, faster training convergence speeds, and potentially better sample complexity.
- We provide a theoretical framework for variants of MoEfication, highlighting conditions when activation sparse experts could become more robust to input noise.
- We verify the relevance of our theoretical setup with experiments on standard large language model benchmarks.

## 2 RELATED WORK

**Theoretical understanding of MoE** The theoretical underpinnings of Mixture-of-Experts (MoE) models have gained increasing attention. Early investigations, such as Nguyen et al. (2016), focused on the approximation capacity of MoEs, establishing a universal approximation theorem for models employing softmax gating functions and linear experts. Subsequent research has focused on the convergence properties of MoE models in regression tasks. For instance, Ho et al. (2022) established the convergence rate of the maximum likelihood estimator (MLE) for MoEs with covariate-free gating functions, drawing connections between expert functions and a class of partial differential equations. Leveraging novel Voronoi-based losses, a series of studies further investigated MLE parameter estimation for Gaussian MoEs across diverse gating functions, including softmax (Nguyen et al., 2023c), Gaussian density (Nguyen et al., 2024b), top-K sparse softmax (Nguyen et al., 2023b), and cosine gating (Nguyen et al., 2024a). In the context of classification, Nguyen et al. (2023a) established convergence rates for density estimation and parameter estimation for MoE models with softmax gating and multinomial logistic experts. Progress has also been made in understanding

MoEs with more complex experts, such as deep neural networks. Chen et al. (2022) considered an MoE model with softmax gating and two-layer CNN experts, demonstrating its superiority over single-expert models due to the router's ability to learn data cluster structures and assign inputs to the most suitable expert. Further research on patch-level routing in MoE (pMoE) with linear gating showed that pMoE outperforms its single-expert two-layer CNN counterparts in terms of sample complexity, even with a smaller model capacity (Chowdhury et al., 2023). These works primarily focus on the routing mechanism, which is difficult to learn from scratch. In contrast, a primary advantage of MoEfication is that it can search for given activation sparsity patterns in a teacher model, which reduces the routing to a clustering problem. Accordingly, the learning challenge in our theoretical setting is induced by handling feature noise.

**Robustness of MoE**  The robustness of Mixture-of-Experts is an increasingly important area of study. For example, Puigcerver et al. (2022) advanced the theoretical understanding of adversarial robustness by proving that MoEs have significantly smaller Lipschitz constants than their dense counterparts. Zhang et al. (2023) proposed a robust training method for CNN-based MoE models that uses alternating adversarial training on both the gating network and the experts, significantly improving stability against attacks. Zhang et al. (2025) incorporated robustness into MoE models while maintaining high accuracy by the introduction of a dual-model strategy, which combines standard and robust MoEs to achieve a balanced trade-off between robustness and accuracy. Li et al. (2022a) suggests that sparse MoE models are inherently strong domain-generalizable learners. By assigning distinct knowledge domains to different experts, MoE architectures can naturally handle domain shifts more effectively than monolithic dense models. This aligns with our paper's claim that the structural properties of MoE contribute fundamentally to its enhanced robustness, yet, in our case rebustness with respect to feature noise.

**Activation sparsity**  Our theoretical setting is motivated by work that exploits activation sparsity to convert pre-trained LLMs into MoEs. Activation sparsity, characterized by a significant portion of zero-valued entries in a model's hidden states, naturally emerges in the intermediate states of traditional ReLU-based transformers (Li et al., 2022b). Early work utilized this sparsity to accelerate LLM inference by optimizing data transfer, such as avoiding weight channel transfers to GPU registers (Liu et al., 2023) or reducing weight movement during CPU offloading (Song et al., 2024b). However, modern LLMs often employ non-ReLU activations, such as SwiGLU (Shazeer, 2020) and GeGLU (Team et al., 2024). Consequently, recent research has focused on inducing activation sparsity in these newer architectures. Strategies include replacing activations like SiLU or GeLU with ReLU variants (e.g., standard ReLU or squared ReLU) combined with continued pretraining or regularization (Mirzadeh et al., 2023; Zhang et al., 2024), and developing specialized pruning methods. For instance, CATS (Lee et al., 2024) enables training-free partial sparsity in SiLU models via magnitude pruning on gate outputs, while TEAL (Liu et al., 2024b) achieves 40-50 percent model-wide sparsity by pruning low-magnitude activations based on observed zero-mean unimodal distributions in LLaMA-style models.

**Error-in-variable regression / noisy features**  Our analysis of feature noise is related to error-in-variable (EIV) regression. EIV, also termed measurement error, addresses scenarios where independent variables (predictors or features) are subject to noise or measurement errors. Despite being common in practice, it presents a departure from standard regression which assumes predictors are exact (Bickel & Ritov, 1987), which is theoretically better tractable. Ignoring these errors can lead to biased estimates, such as attenuation bias in simple linear regression (Frost & Thompson, 2000). Recent research integrates EIV concepts with modern machine learning phenomena observed in high-dimensional settings. For example, it is demonstrated in (Kausik et al., 2023) that test error curves can exhibit double descent patterns even under distribution shift, with feature noise acting as an implicit regularizer.

## 3  PROBLEM SETTING

**Notations**  We use lower-case and upper-case letters (e.g., $x$ and $X$) to represent vectors and matrices respectively. The entries of the vectors and matrices are denoted by $x_i$ and $X_{ij}$, respectively. For a given matrix $X \in \mathbb{R}^{n \times d}$, we denote its singular values in a decreasing order $\lambda_{1,n} \geq \lambda_{2,n} \geq \cdots \geq \lambda_{\min(d,n),n} \geq 0$. Their limits, as $n, d \to \infty$, are denoted as $\lim_{n \to \infty} \lambda_{i,n} = \lambda_i$.

Assume that $\lambda_{\min}(X)$ denotes the minimum singular value of matrix $X$. The aspect ratio $c_n$ for the above matrix is defined by $c_n := d/n$ and the corresponding limit is $\lim_{n \to \infty} c_n = c$. We denote the univariate and multivariate Gaussian distribution by $\mathcal{N}(\mu, \sigma^2)$ and $\mathcal{N}(\mu, \Sigma)$ respectively. We use $I_n$ to denote the identity matrix of $n$ dimension. Assume $()^+$ to be the Moore-Penrose inverse of the matrix. Let $\odot$ denote the element-wise product.

**Problem formulation**   To theoretically investigate the advantages of MoEs, particularly in the context of activation sparsity and resilience to noise, we consider a simplified linear model. This model aims to capture the essence of how MoEs might effectively utilize underlying sparse structures within data or activations, even when these structures are obscured by feature noise. This setting provides a tractable framework for comparing dense versus sparse (MoE-like) estimation strategies.

## 3.1 WHY LINEAR MODELS?

Our adoption of a simplified linear model is a deliberate methodological choice, enabling a tractable yet insightful analysis for several key reasons.

**Theoretical tractability.** From a theoretical standpoint, starting with a linear model is a common and established practice in deep learning theory and has led to impactful insights, including the discovery of the double descent phenomenon (Belkin et al., 2019), implicit biases (Jacobs et al., 2025), Neural Tangent Kernels (Jacot et al., 2018), or mean field theory (Sirignano & Spiliopoulos, 2019). It allows us to isolate the core mechanism—how activation sparsity confers robustness to feature noise—in a theoretically manageable manner. This approach aligns with the NTK (Jacot et al., 2018) viewpoint, where in certain regimes, deep networks can be well-approximated by linear functions, making our analysis relevant to the broader study of deep learning optimization.

**Relevance to finetuning.** Furthermore, our linear setup finds strong parallels in the finetuning of LLMs through the lens of the *linear representation hypothesis*, which posits that features within LLMs are often linearly represented (Mikolov et al., 2013). This has given rise to *linear probing*, a standard technique where simple linear models are trained on the internal activations of a frozen LLM to assess the information they contain (Gurnee et al., 2023; Jiao et al., 2023). In this context, our MoE framework can be viewed as a system of specialized linear probes, where the gating mechanism routes an input to the most appropriate probe (expert) based on its internal features. This perspective grounds our work in the practical and widely-used methods for interpreting and fine-tuning large-scale models.

**Empirical validation in non-linear settings.** To ensure our findings generalize beyond the linear case, we conducted controlled experiments on a synthetic dataset using a non-linear, two-layer MoE network with ReLU activations. We evaluated both regression and classification tasks under noisy conditions. The results, detailed in Appendix B.3, consistently show that the non-linear MoE models are more robust to noise than their dense counterparts (see Table 2 and Table 3). This suggests that the robustness gains we identify stem fundamentally from the modular structure of the features, an insight that holds even in more complex, non-linear architectures.

## 3.2 LINEAR MODELS WITH BLOCK STRUCTURE

Let $\beta^\star = [\beta_1^{\star T}, \ldots, \beta_k^{\star T}]^T \in \mathbb{R}^d$ be the ground truth parameter vector, partitioned into $k$ blocks corresponding to $k$ distinct "experts". The output vector $Y = [y_1, \ldots, y_n]^T \in \mathbb{R}^n$, comprising $n$ samples, is generated by:

$$Y = X\beta^\star, \quad X = \begin{bmatrix} X_1 & 0 & \ldots & 0 \\ 0 & X_2 & \ldots & 0 \\ \vdots & \vdots & \ddots & \vdots \\ 0 & 0 & \ldots & X_k \end{bmatrix}, \tag{1}$$

where $X \in \mathbb{R}^{n \times d}$ is the noiseless design matrix. We assume that $X$ possesses a block-diagonal structure, where each $X_i \in \mathbb{R}^{n_i \times d_i}$ (with $\sum n_i = N'$ for some $N'$ samples actually contributing to specific experts, and $n$ rows in $X$ formed by appropriate zero-padding for samples not pertinent to an expert block, or $\sum d_i = d$ if $X_i$ are sub-matrices of features for all $n$ samples). This block-diagonal structure represents an idealized scenario where distinct input features (or latent representations) are

processed by distinct experts. This is analogous to the goal of MoEfication, where one seeks to identify or create such specialized, sparsely activated pathways from a dense network. $X_i$ can be fixed or random; for instance, its elements might be sampled from a Gaussian distribution, reflecting empirical observations in network activations (Liu et al., 2024b).

In practice, we only observe a noisy version of the design matrix, $\bar{X} = X + E$, where $E = [E_{ij}]$ with $E_{ij} \sim \mathcal{N}(0, \sigma^2)$ representing feature noise. This noise can be interpreted not only as direct perturbations to input features but also as a proxy for the interference or lack of clear modularity in observed activations within a dense network before techniques like pruning reveal underlying sparse structures. We still have access to the true output $Y$.

We compare two approaches to estimate $\beta^\star$:

1. **Dense Estimator**: This approach uses the full noisy design matrix $\bar{X}$, analogous to a single, dense model attempting to learn all expert specializations simultaneously. The minimum norm estimator is:

$$\hat{\beta} = \underset{\beta}{argmin} \left\{ ||\beta||_2^2 \Big| \beta \in \underset{\beta}{argmin} \, ||Y - \bar{X}\beta||_2^2 \right\} = \left( \bar{X}^T \bar{X} \right)^+ \bar{X}^T Y$$

2. **Sparse Estimators (MoE-like)**: Assuming a perfect gating mechanism (justified if signal-to-noise is sufficient), we can isolate the estimation for each expert. For the $i$-th expert, using its corresponding data block $Y_i$ (rows of $Y$ corresponding to $X_i$) and noisy features $\bar{X}_i$ (the $i$-th block of $\bar{X}$ corresponding to $X_i$), the estimator is:

$$\hat{\beta}_i = \left( \bar{X}_i^T \bar{X}_i \right)^+ \bar{X}_i^T Y_i = \left( \bar{X}_i^T \bar{X}_i \right)^+ \bar{X}_i^T \left( X_i \beta_i^\star \right)$$

This decomposition into $\hat{\beta}_i$ models the behavior of an MoE where each expert focuses on a specific sub-problem. The challenge of recovering $Y_i$ or $X_i \beta_i^\star$ from noisy inputs $\bar{X}_i$ mirrors the practical scenario where an MoE, derived from a dense model, aims to reconstruct the dense model's (or an idealized target's) behavior using its specialized experts. This perspective connects to knowledge distillation, where the MoE's output (based on $\hat{\beta}_i$) is trained to match the output of a teacher (related to $Y$ or $X\beta^\star$).

Direct analysis of these minimum norm estimators under feature noise is challenging. Therefore, we first analyze their Bayes optimal counterparts, assuming access to the data distribution, to understand their performance limits with infinite data points. Subsequently, we examine the convergence dynamics towards these optimal estimators.

The Bayes optimal regressor $f^\star$ minimizes the mean squared error:

$$f^\star := \underset{f}{argmin} \, \mathbb{E}_{(x,y)\sim\mathcal{D}} \left[ (f(x) - y)^2 \right].$$

Given our multi-expert setup, where data $(x, y)$ might arise from different underlying distributions $\mathcal{D}_i$ for each expert $i \in [k]$, we introduce a latent variable $z$ indicating the active expert:

$$f^\star := \underset{f}{argmin} \sum_{i=1}^{k} \mathbb{P}(z = i) \mathbb{E}_{(x,y)\sim\mathcal{D}_i} \left[ (f(x) - y)^2 \big| z = i \right]. \qquad (2)$$

For linear regressors $f(x) = x^T \beta$, the Bayes optimal estimators for the dense case, $\beta_{Dense}^{Bayes}$, and for the $i$-th sparse expert, $\beta_{Sparse,i}^{Bayes}$, are derived from minimizing this objective. The derived Bayes optimal estimators are:

$$\beta_{Dense}^{Bayes} = \begin{bmatrix} p_1(p_1\Sigma_1 + \sigma^2 I)^{-1}\Sigma_1\beta_1^\star \\ \vdots \\ p_k(p_k\Sigma_k + \sigma^2 I)^{-1}\Sigma_k\beta_k^\star \end{bmatrix}, \quad \beta_{Sparse,i}^{Bayes} = (\Sigma_i + \sigma^2 I)^{-1}\Sigma_i\beta_i^\star, \, i = 1, \ldots, k. \quad (3)$$

where $\Sigma_i$ is the covariance matrix of the noiseless features $x$ pertinent to expert $i$, $p_i = \mathbb{P}(z = i)$ is the probability of selecting expert $i$, and $\sigma^2$ relates to the variance of the feature noise $E$. These forms arise from a Bayesian linear regression perspective under Gaussian assumptions for features and noise.

# 4 MAIN RESULTS: MoEs CAN HANDLE FEATURE NOISE

This section presents our main theoretical findings, establishing the advantages of MoE-like sparse estimators over dense estimators in the context of feature noise. We analyze generalization error, robustness to perturbations, training convergence speed and sample complexity for the excess risks. All proofs are provided in the appendix.

## 4.1 ROUTING SIMPLIFIES TO CLUSTERING

Traditional MoE models face the challenge of learning a router, which reduces to clustering in the context of MoEfication or a clear block structure. For that reason, a Bayes optimal estimate of the router can be regarded as nearly perfect, as we establish in the following.

**Decoupled training as a supervised task.** Unlike traditional MoEs where the router and experts are trained jointly from scratch in a complex optimization landscape, our approach first identifies expert structures by clustering neurons in a pre-trained dense model. This initial step provides pre-defined, ground-truth "labels" for each data point, indicating which expert it belongs to. Consequently, training the router is no longer a difficult joint optimization problem. Instead, it becomes a standard, well-posed supervised classification task: learning to predict the correct expert for a given input. This task is significantly more tractable.

**Theoretical guarantees.** The geometric separation of data in our modular model makes this classification task straightforward. We provide a theoretical analysis showing that a simple and efficient classifier can achieve near-perfect routing accuracy with a practical number of samples.

**Theorem 1** (Informal). *Under the assumption of a modular data structure 1, a Quadratic Discriminant Analysis (QDA) based router achieves an excess risk of less than $\epsilon$ with high probability for $n \geq \mathcal{O}(poly(d, \log(1/\delta)))$ samples.*

This result demonstrates that achieving a nearly perfect router is theoretically feasible within our framework. The formal statement of Theorem 1 and its complete proof are provided in Appendix A.7. This decouples the analysis of expert performance from the challenge of routing, allowing us to gain clearer insights into the inherent strengths of the expert structure itself.

## 4.2 GENERALIZATION

The following theorem compares the generalization errors of the Bayes optimal estimators for the dense and sparse cases, as defined in Eq. (3).

**Theorem 2.** *Consider the linear model defined in Eq. (1) and the Bayes optimal estimators $\beta_{Dense}^{Bayes}$ and $\beta_{Sparse,i}^{Bayes}$ Eq. (3). The corresponding generalization errors are:*

$$\mathcal{R}(\beta_{Sparse}^{Bayes}) = \sum_{i=1}^{k} p_i \sigma^2 \beta_i^{\star T} \Sigma_i (\Sigma_i + \sigma^2 I)^{-1} \beta_i^{\star}, \ \mathcal{R}(\beta_{Dense}^{Bayes}) = \sum_{i=1}^{k} p_i \sigma^2 \beta_i^{\star T} \Sigma_i (p_i \Sigma_i + \sigma^2 I)^{-1} \beta_i^{\star}.$$

A key implication of Theorem 2 is that sparse estimators achieve better generalization performance than the dense estimator. Since $0 < p_i \leq 1$ and $\Sigma_i$ is positive semi-definite, $p_i \Sigma_i \preceq \Sigma_i$. Consequently, $p_i \Sigma_i + \sigma^2 I \preceq \Sigma_i + \sigma^2 I$. For positive definite matrices $A \preceq B$, it holds that $B^{-1} \preceq A^{-1}$. Thus, $(\Sigma_i + \sigma^2 I)^{-1} \preceq (p_i \Sigma_i + \sigma^2 I)^{-1}$. This means each term in the sum for $\mathcal{R}(\beta_{Sparse}^{Bayes})$ is less than or equal to the corresponding term for $\mathcal{R}(\beta_{Dense}^{Bayes})$, leading to $\mathcal{R}(\beta_{Sparse}^{Bayes}) \leq \mathcal{R}(\beta_{Dense}^{Bayes})$.

## 4.3 ROBUSTNESS TO INPUT NOISE

Theorem 2 also provides a basis for understanding out-of-distribution generalization, or robustness to input perturbations. We consider two scenarios for perturbations:

1) **Perturbations not affecting routing:** The router assigns perturbed inputs to the correct experts.

2) **Perturbations causing mis-routing:** The router assigns perturbed inputs to incorrect experts.

The following theorem addresses the first scenario.

**Theorem 3.** *Assume input perturbations are modeled as Gaussian noise with variance $\sigma_o^2$, and the router consistently makes correct expert assignments. The generalization errors under such perturbations for the dense and sparse estimators are:*

$$\mathcal{R}_{Dense}(\sigma_o^2) = \sum_{i=1}^{k} p_i \sigma^2 \beta_i^{\star T} \Sigma_i (p_i \Sigma_i + \sigma^2 I)^{-1} \beta_i^{\star} + \sum_{i=1}^{k} p_i^2 (\sigma_o^2 - \sigma^2) \beta_i^{\star T} \Sigma_i (p_i \Sigma_i + \sigma^2 I)^{-2} \Sigma_i \beta_i^{\star},$$

$$\mathcal{R}_{Sparse}(\sigma_o^2) = \sum_{i=1}^{k} p_i \sigma^2 \beta_i^{\star T} \Sigma_i (\Sigma_i + \sigma^2 I)^{-1} \beta_i^{\star} + \sum_{i=1}^{k} p_i (\sigma_o^2 - \sigma^2) \beta_i^{\star T} \Sigma_i (\Sigma_i + \sigma^2 I)^{-2} \Sigma_i \beta_i^{\star}.$$

Sparse estimators can exhibit improved robustness (i.e., $\mathcal{R}_{Sparse}(\sigma_o^2) \leq \mathcal{R}_{Dense}(\sigma_o^2)$) under these conditions, particularly when $\sigma_o^2 > \sigma^2$. A sufficient condition for this, building upon the baseline advantage from Theorem 2, is if the impact of the additional error term $(\sigma_o^2 - \sigma^2)$ is less detrimental for the sparse estimators. For instance, if $\lambda_{\min}(\Sigma_i) > 4\sigma^2$ (indicating a sufficiently high signal-to-noise ratio for each expert's features) and $\sigma_o^2 > \sigma^2$, sparse estimators are favored.

However, if perturbations are large enough to cause mis-routing, the situation can change. The following theorem considers a specific mis-routing scenario.

**Theorem 4.** *Consider an input originally intended for expert $i$, $x_i \sim \mathcal{N}(0, \Sigma_i)$, but a perturbation of the form $\eta x_j$ (where $x_j \sim \mathcal{N}(0, \Sigma_j)$, $\eta > 1$) is introduced, causing the router to select expert $j$ with high probability. Let the overall noisy input be composite, denoted abstractly as $[0, \ldots, 0, x_i^T, 0, \ldots, 0, \eta x_j^T, 0, \ldots, 0]^T + e$, where $e \sim \mathcal{N}(0, \sigma^2 I_d)$. The generalization errors for the dense estimator:*

$$\mathcal{R}_{Dense}^{mis\text{-}route} = \eta^2 p_j \beta_j^{\star T} \Sigma_j (p_j \Sigma_j + \sigma^2 I_d)^{-1} \Sigma_j \beta_j^{\star} + \sigma^2 \eta^2 (p_j^2 - p_j) \beta_j^{\star T} \Sigma_j (p_j \Sigma_j + \sigma^2 I_d)^{-2} \Sigma_j \beta_j^{\star}$$

$$- p_i \beta_i^{\star T} \Sigma_i (p_i \Sigma_i + \sigma^2 I_d)^{-1} \Sigma_i \beta_i^{\star} + \sigma^2 \sum_{r \neq i,j}^{k} p_r^2 \beta_r^{\star T} \Sigma_r (p_r \Sigma_r + \sigma^2 I_d)^{-2} \Sigma_j \beta_r^{\star}$$

*And for the sparse (MoE-like) estimator:* $\mathcal{R}_{Sparse}^{mis\text{-}route} = \eta^2 \beta_j^{\star T} \Sigma_j (\Sigma_j + \sigma^2 I_d)^{-1} \Sigma_j \beta_j^{\star}$.

The expressions in Theorem 4 are complex to compare directly. However, they illustrate that the performance dynamics can shift under mis-routing. In certain situations, such as when $p_r = 0$ for $r \neq i, j$ (i.e., only experts $i$ and $j$ have non-zero selection probabilities), the dense estimator might handle such specific perturbations more effectively than a sparse estimator that is forced to use a highly specialized but incorrect expert. This suggests a trade-off: specialized experts excel when routing is correct but can be detrimental if routing fails significantly.

### 4.4 CONVERGENCE SPEED

Next, we analyze the convergence dynamics of gradient descent algorithms for learning the dense and sparse estimators. We make the following simplifying assumptions: For the design matrix $X$:

    i) Each $X_i$ is fixed and of size $\frac{n}{k} \times \frac{d}{k}$.

    ii) The asymptotic aspect ratio $c = \lim_{n,d \to \infty} d/n > 1$. The singular values $\lambda_{ij,n}$ of $X_i$ (for $j = 1, \ldots, n/k$) converge to $\lambda_{ij}$ as $n, d \to \infty$.

    iii) For all $i, j$, $\lambda_{ij} > \sqrt{c}\sigma^2$. This condition relates to signal strength versus noise level.

Under these assumptions, the following theorem characterizes the asymptotic convergence rates.

**Theorem 5.** *Under Assumption 4.4, the convergence rate (error reduction factor per iteration) for the $i$-th sparse estimator and dense estimator using gradient descent is given by:*

$$\rho_{Sparse,i} = \left( 1 - \frac{\lambda_{i1}^2 \left( \sigma^2 + \lambda_{i\frac{n}{k}}^2 \right) \left( c\sigma^2 + \lambda_{i\frac{n}{k}}^2 \right)}{\lambda_{i\frac{n}{k}}^2 \left( \sigma^2 + \lambda_{i1}^2 \right) \left( c\sigma^2 + \lambda_{i1}^2 \right)} \right),$$

$$\rho_{Dense} = \left( 1 - \frac{\max_j \lambda_{j1}^2 \left( \sigma^2 + \min_l \lambda_{l\frac{n}{k}}^2 \right) \left( c\sigma^2 + \min_l \lambda_{l\frac{n}{k}}^2 \right)}{\min_l \lambda_{l\frac{n}{k}}^2 \left( \sigma^2 + \max_j \lambda_{j1}^2 \right) \left( c\sigma^2 + \max_j \lambda_{j1}^2 \right)} \right).$$

*Typically, $\rho_{Sparse,i} \leq \rho_{Dense}$, which implies faster convergence for sparse estimators. At most one sparse estimator (the one whose singular value spectrum matches the terms defining $\rho_{Dense}$) will have a convergence rate equal to that of the dense estimator; others will generally converge faster.*

## 4.5 SAMPLE EFFICIENCY

Finally, we consider the sample complexity of the excess risk for the dense and sparse estimators. While a full theoretical derivation is beyond the scope of this paper and left for further exploration, we posit a hypothesis based on empirical observations (e.g., Figure 2) and general statistical learning principles: *Sparse estimators achieve lower excess risk for a given sample size, or equivalently, require fewer samples to reach a target excess risk, compared to dense estimators. This implies a better sample complexity for MoEs.*

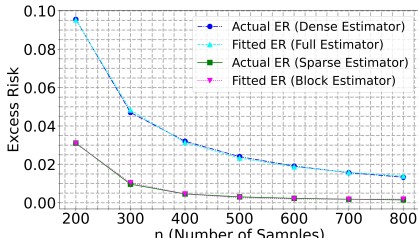

This hypothesis is supported by intuitive arguments from a bias-variance perspective. For a given input, the ground truth parameter vector effectively lies in a lower-dimensional subspace (e.g., $s$-dimensional) for a sparse expert, compared to the full $d$-dimensional space for the dense estimator ($s < d$). The dense estimator's attempt to learn signals in the extraneous $d - s$ dimensions can introduce larger bias. Regarding variance, sparse estimators are affected by noise primarily within their relevant $s$-dimensional subspace, while the dense estimator's variance is influenced by noise in the full $d$-dimensional space, potentially leading to higher variance. Experimental results, such as those shown in Figure 3, corroborate this conjectured advantage. Our analysis, fitting excess

Figure 2: Fitting the curve of the excess risks with respect to $n$ only. The result shows that the excess risk for both dense and sparse estimators is of order $n^{-2}$, with different scalars only.

risk solely against $n$, shows that both dense and sparse estimators achieve an excess risk of order $O(n^{-2})$, with variation only in their scalar multipliers. This similarity renders comparison methods that depend on distinguishing estimators by such orders inapplicable. Moreover, the complex functional form of the theoretical result already for a noiseless design matrix (Bartlett et al., 2020) prevents its estimation via curve fitting with feature noise. Experimental results for various specialized settings are provided in the appendix.

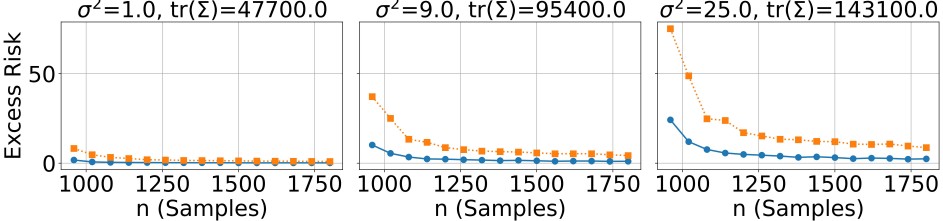

Figure 3: Comparison of excess risks. The figure plots the excess risk for 'Sparse Estimators' against a 'Dense Estimator' as a function of the sample size $n$. We assume a dimensionality $p = 900$, number of experts $k = 12$ and that each input has the same size. The covariance matrix of $X_i$ is a Toeplitz matrix. In the legend, the 'Sparse Estimators' are depicted by the solid line with 'o' markers, while the 'Dense Estimator' is shown with a dotted line and 's' markers.

## 5 EXPERIMENTS

**Modular structures** First, we provide empirical support for a key assumption in our theoretical analysis: a block-diagonal feature matrix structure module feature noise. Figure 1 demonstrates that the latent feature space of large language models (LLMs) can exhibit a similar modular, block-diagonal structure. Additional visualizations are provided in the appendix. Pretrained LLMs learn highly transferable features, requiring only a few epochs of finetuning to perform well on downstream tasks. Since these features are transferable, they can be investigated to better understand relatively general properties of activations. Specifically, we analyze the activations of intermediate layers of the Llama-2 7B model (Touvron et al., 2023) for tokens of the WikiText2 dataset (Merity et al., 2016). To highlight the block structure, we employ the recently proposed activation sparsity

method TEAL (Liu et al., 2024b) that removes low-magnitude activations. While these pruned activations are not necessarily noise, setting them to zero has only little impact on model performance.

**Robustness to noise** According to our theory, we expect activation sparse MoEs to exhibit higher robustness to feature noise. To understand whether this insight could transfer into practice, we examine the robustness of dense and sparsely activated models under realistic noise conditions. We use MoEfication to generate the sparsely activated model, following the approach of (Zhang et al., 2021). Specifically, we apply MoEfication to the T5-base model (Raffel et al., 2020) and evaluate performance on the development set of the SST-2 dataset (Socher et al., 2013). The acti-

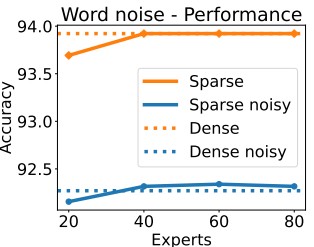 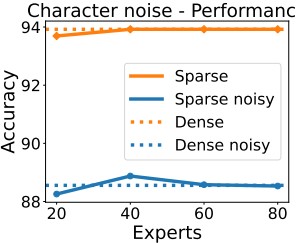

Figure 4: Robustness to noise. The figure shows the mean performance of dense and activation sparse T5-base models. Noise has been applied to the inputs of the SST2 dataset over 5 seeds. (Left) Word noise, (right) Character noise.

vations are partitioned into 96 experts, consistent with the original MoEfication setup. To simulate noise, we adopt the perturbation schemes from (Pan et al., 2024), introducing both word-level and character-level noise. Word noise is applied by randomly swapping adjacent words up to two times per sentence, while character noise involves inserting keyboard-based spelling errors in two words per sentence, affecting at most one character per word. As shown in Fig. **??**&4, sparsely activated models experience a smaller drop in accuracy at intermediate sparsity levels when exposed to input noise compared to dense models. Moreover, sparse models consistently outperform their dense counterparts under noisy conditions at these sparsity levels. These results support our finding that sparse estimators can offer improved robustness to noise under realistic conditions and provide a novel insight into the potential benefits of MoEfication: Not only can it lead to inference speed-ups, but also to higher robustness to noise.

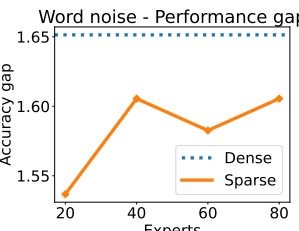 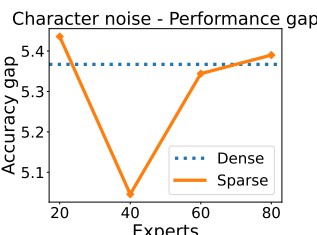

Figure 5: As Fig. 4, but the performance gap resulting from noisy inputs is shown.

## 6 CONCLUSIONS

This paper unmasks a novel mechanism underpinning the success of MoE models, demonstrating that MoEs can outperform their dense counterparts by effectively utilizing activation sparsity in the presence of feature noise, even without an increase in total parameter size. This inherent structural advantage translates into improved generalization, enhanced robustness to input perturbations, faster training convergence, and better sample complexity. Our analysis lays thus a theoretical foundation for MoEfication, which utilizes hidden block structures of single expert teacher models. Experiments on LLMs highlight the practical value of our theoretical insights. Our insights could have a significant broader impact, particularly on the development of more computationally efficient LLMs. By understanding how to leverage sparsity, MoE architectures can reduce inference costs, thereby improving the accessibility of large models and potentially reducing their environmental footprint.

## REPRODUCIBILITY STATEMENT

To ensure the reproducibility of our research, we provide comprehensive documentation of our theoretical and empirical contributions. The complete proofs for all theoretical results presented in this paper can be found in the appendix. Furthermore, a detailed description of our experimental setup is also provided in the appendix. The full source code for conducting the experiments is included as part of the supplementary material.

## LLM USAGE DISCLOSURE

We acknowledge the use of Large Language Models (LLMs) in the preparation of this manuscript. Our use of LLMs was for the following purposes:

- To revise and enhance the language and clarity of the full text.
- To generate portions of the experimental code.

All authors have reviewed, edited, and take full responsibility for the final content of this paper, including the accuracy and validity of the LLM-generated code and text.

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

## LLM USAGE DISCLOSURE

We acknowledge the use of Large Language Models (LLMs) in the preparation of this manuscript. Our use of LLMs was for the following purposes:

- To revise and enhance the language and clarity of the full text.
- To generate portions of the experimental code.

All authors have reviewed, edited, and take full responsibility for the final content of this paper, including the accuracy and validity of the LLM-generated code and text.

## A    PROOFS AND FURTHER ANALYSIS

### A.1    DERIVATION OF BAYES OPTIMAL ESTIMATORS

To derive the Bayes optimal estimators for both the dense and sparse cases, we consider the scenario with infinite data samples where we know the distribution of the covariates, the output, and the feature noise. Under these conditions, we can compute the generalization error as:

$$
\begin{aligned}
\mathcal{R}(\hat{\beta}) &= \sum_{i=1}^{k} \mathbb{P}(z=i)\mathbb{E}_{(x,y)\sim\mathcal{D}_i}\left[(f(x)-y)^2 | z=i\right] \\
&= \sum_{i=1}^{k} p_i \mathbb{E}_{x\sim D_i, e}\left[((x+e)^T\hat{\beta} - x^T\beta_i^\star)^2\right] \\
&= \sum_{i=1}^{k} p_i \mathbb{E}_{x\sim D_i, e}\left[(\beta_i^\star)^T xx^T \beta_i^\star + \hat{\beta}^T xx^T \hat{\beta} + \hat{\beta}^T ee^T \hat{\beta} - 2\hat{\beta}^T xx^T (\beta_i^\star)^T\right] \\
&= \sum_{i=1}^{k} p_i \left((\beta_i^\star)^T \Sigma_i \beta_i^\star + \hat{\beta}_i^T \Sigma_i \hat{\beta}_i + \sigma^2 \hat{\beta}^T \hat{\beta} - 2\hat{\beta}_i^T \Sigma_i (\beta_i^\star)^T\right)
\end{aligned}
\tag{4}
$$

The third equality follows from the independence of the covariate $x$ and feature noise $e$, which makes the expectation of their cross terms equal to zero. The final equality uses the covariance matrices of $x$ and $e$.

The Bayes optimal estimator minimizes the generalization error, which requires:

$$
\nabla_{\hat{\beta}}\mathcal{R}(\hat{\beta})\bigg|_{\hat{\beta}=\beta_{Dense}^{Bayes}} = 0
$$

Substituting Equation (4) and solving this optimization problem yields the form of $\beta_{Dense}^{Bayes}$. The Bayes sparse estimators can be obtained using the same approach.

### A.2    PROOF OF THEOREM 2

The generalization error of the Bayes dense estimator is:

$$
\begin{aligned}
\mathcal{R}(\beta_{Dense}^{Bayes}) &= \sum_{i=1}^{k} p_i \left((\beta_i^\star)^T \Sigma_i \beta_i^\star + (\beta_{Dense,i}^{Bayes})^T \Sigma_i \beta_{Dense,i}^{Bayes} + \sigma^2 (\beta_{Dense}^{Bayes})^T \beta_{Dense}^{Bayes} - 2(\beta_{Dense,i}^{Bayes})^T \Sigma_i \beta_i^\star\right) \\
&= \sum_{i=1}^{k} p_i \sigma^2 (\beta_i^\star)^T \Sigma_i (p_i \Sigma_i + \sigma^2 I)^{-1} \beta_i^\star
\end{aligned}
$$

This result is obtained by substituting $\beta_{Dense}^{Bayes}$ into Equation (4). The proof for the generalization error of sparse estimators follows similar derivations and is therefore omitted.

### A.3 PROOF OF THEOREM 3

The process to obtain the robustness under correct routing for both the sparse estimators and dense estimator is similar, so we take the dense estimator as an example. The generalization error of the dense estimator is:

$$
\mathcal{R}_{Dense}(\sigma_o^2) = \sum_{i=1}^{k} p_i \mathbb{E}_{x \sim D_i, e' \sim \mathcal{N}(0, \sigma_o^2 I_d)} \Big[ (\beta_i^\star)^T x x^T \beta_i^\star + (\beta_{Dense,i}^{Bayes})^T x x^T \beta_{Dense,i}^{Bayes} + (\beta_{Dense}^{Bayes})^T e' e'^T \beta_{Dense}^{Bayes}
$$

$$
- 2(\beta_{Dense,i}^{Bayes})^T x x^T \beta_i^\star \Big]
$$

$$
= \sum_{i=1}^{k} p_i \left( (\beta_i^\star)^T \Sigma_i \beta_i^\star + (\beta_{Dense,i}^{Bayes})^T \Sigma_i \beta_{Dense,i}^{Bayes} + \sigma_o^2 (\beta_{Dense}^{Bayes})^T \beta_{Dense}^{Bayes} - 2(\beta_{Dense,i}^{Bayes})^T \Sigma_i \beta_i^\star \right)
$$

$$
= \sum_{i=1}^{k} p_i \sigma^2 (\beta_i^\star)^T \Sigma_i (p_i \Sigma_i + \sigma^2 I)^{-1} \beta_i^\star + \sum_{i=1}^{k} p_i^2 (\sigma_o^2 - \sigma^2)(\beta_i^\star)^T \Sigma_i (p_i \Sigma_i + \sigma^2 I)^{-2} \Sigma_i \beta_i^\star
$$

### A.4 PROOF OF THEOREM 4

In the mis-routing case, under the specific perturbation, the result for sparse estimator is just $\eta^2$ scaled of the generalization error in Theorem 2. The generalization error for the input $[0, \ldots, 0, x_i^T, 0, \ldots, 0, \eta x_j^T, 0, \ldots, 0]^T + e$ is:

$$
\mathcal{R}_{Dense}^{\text{mis-route}} = (\beta_i^\star)^T \Sigma_i \beta_i^\star + (\beta_{Dense,i}^{Bayes})^T \Sigma_i \beta_{Dense,i}^{Bayes} + \eta^2 (\beta_{Dense,j}^{Bayes})^T \Sigma_j \beta_{Dense,j}^{Bayes}
$$

$$
+ \sigma^2 (\beta_{Dense}^{Bayes})^T \beta_{Dense}^{Bayes} - 2(\beta_{Dense,i}^{Bayes})^T \Sigma_i \beta_i^\star
$$

$$
= \eta^2 p_j \beta_j^{\star T} \Sigma_j (p_j \Sigma_j + \sigma^2 I_d)^{-1} \Sigma_j \beta_j^\star + \sigma^2 \eta^2 (p_j^2 - p_j) \beta_j^{\star T} \Sigma_j (p_j \Sigma_j + \sigma^2 I_d)^{-2} \Sigma_j \beta_j^\star
$$

$$
- p_i \beta_i^{\star T} \Sigma_i (p_i \Sigma_i + \sigma^2 I_d)^{-1} \Sigma_i \beta_i^\star + \sigma^2 \sum_{r \neq i,j}^{k} p_r^2 \beta_r^{\star T} \Sigma_r (p_r \Sigma_r + \sigma^2 I_d)^{-2} \Sigma_j \beta_r^\star
$$

### A.5 PROOF OF THEOREM 5

We first establish that the convergence rate of the least squares estimator is related to the condition number of the design matrix $X$. Using gradient descent, the update step follows:

$$
\hat{\beta}_{t+1} = \hat{\beta}_t - \eta_t \left( X^T X \hat{\beta}_t - X^T y \right) \tag{5}
$$

where $\eta_t$ is the step size.

Let the singular value decomposition of the design matrix $X$ be $U \Sigma V^T$, and choose the step size $\eta_t = \frac{1}{\lambda_1^2}$ for all $t$, where $\lambda_1$ is the maximal singular value of $X$. Then the residual follows the dynamics:

$$
X \hat{\beta}_{t+1} - y = X \hat{\beta}_t - y - \eta_t X \left( X^T X \hat{\beta}_t - X^T y \right)
$$

$$
= X \hat{\beta}_t - y - \eta_t X X^T \left( X \hat{\beta}_t - y \right)
$$

$$
= \left( I_n - \eta_t X X^T \right) \left( X \hat{\beta}_t - y \right)
$$

$$
= U \text{diag} \left( 1 - \frac{\sigma_1^2}{\sigma_1^2}, \ldots, 1 - \frac{\sigma_d^2}{\sigma_1^2} \right) U^T \left( X \hat{\beta}_t - y \right)
$$

$$
= U \text{diag} \left( \left( 1 - \frac{\sigma_1^2}{\sigma_1^2} \right)^{t+1}, \ldots, \left( 1 - \frac{\sigma_d^2}{\sigma_1^2} \right)^{t+1} \right) U^T \left( X \hat{\beta}_0 - y \right)
$$

Therefore, the convergence rate depends on $1 - \frac{\lambda_d^2}{\lambda_1^2}$. For the design matrices $\bar{X}_i = X_i + E_i$ (where $i = 1, \ldots, k$) and $\bar{X}$, the convergence rates depend on $1 - \frac{\bar{\lambda}_{\frac{d}{k}}^2}{\bar{\lambda}_{i1}^2}$ and $1 - \frac{\bar{\lambda}_d^2}{\bar{\lambda}_1^2}$, respectively.

Based on Theorem 6 in Loubaton & Vallet (2011), the perturbed singular values $\bar{\lambda}_i^2$ and the non-perturbed singular values $\lambda_i^2$ satisfy the following relationship as $n \to \infty$:

$$\bar{\lambda}_i^2 \to \begin{cases} \frac{(\sigma^2 + \lambda_i^2)(c\sigma^2 + \lambda_i^2)}{\lambda_i^2}, & \text{if } \lambda_i^2 > \sqrt{c}\sigma^2 \\ \sigma^2(1 + \sqrt{c})^2, & \text{otherwise} \end{cases}$$

Substituting this result into the convergence speed analysis yields Theorem 5.

### A.6    ANALYSIS OF SAMPLE COMPLEXITY OF MINIMUM NORM ESTIMATORS

As mentioned in the main text, analyzing the sample complexity of minimum norm estimators presents significant challenges. The difficulty arises not only because the sample complexities of both sparse and dense estimators have the same order, but also because analyzing the excess risk of each estimator individually is inherently difficult.

The excess risk is the difference between the test error and the irreducible risk (the Bayes optimal error). We first examine the test error structure since we already have the Bayes optimal risk.

The bias and variance decomposition of the test error is:

$$\mathbb{E}_{x,e}[(x+e)^T\hat{\beta} - x^T\beta]^2 = \mathbb{E}_{x,e}[x^T(\hat{\beta} - \beta)]^2 + \mathbb{E}_e(e^T\hat{\beta})^2$$

where the first term represents the bias and the second term represents the variance.

#### A.6.1    BIAS ANALYSIS

Since $\bar{X}$ is full rank, we have:

$$\hat{\beta} - \beta^* = -(\bar{X}^T\bar{X})^+\bar{X}^T E\beta^*$$

where $\bar{X} = X + E$. The bias term becomes:

$$\begin{aligned} \mathbb{E}_x[x^T(\hat{\beta} - \beta^*)]^2 &= (\hat{\beta} - \beta^*)^T\mathbb{E}_x[xx^T](\hat{\beta} - \beta^*) \\ &= (\hat{\beta} - \beta^*)^T\Sigma(\hat{\beta} - \beta^*) \\ &= (\beta^*)^T E^T\bar{X}(\bar{X}^T\bar{X})^+\Sigma(\bar{X}^T\bar{X})^+\bar{X}^T E\beta^* \end{aligned}$$

This equality holds for both sparse and dense estimators. However, this expression is difficult to analyze because $E$ and $\bar{X}$ are highly correlated.

#### A.6.2    VARIANCE ANALYSIS

For the $i$-th sparse estimator, we assume that the diagonal block $X_i$ is sampled from a Gaussian distribution $\mathcal{N}(0, \Sigma_i)$, and the covariance matrix has eigenvalue decomposition $\Sigma_i = U_i\Lambda_i U_i^T$, where $\Lambda_i = \text{diag}\{\lambda_{i1}, \ldots, \lambda_{i,n/k}\}$.

$$\begin{aligned} \text{Var}(\hat{\beta}) &= \sigma^2\text{tr}\left[\left(\bar{X}_i^T\bar{X}_i\right)^+ \bar{X}_i^T X_i \beta_i^\star(\beta_i^\star)^T X_i^T\bar{X}_i \left(\bar{X}_i^T\bar{X}_i\right)^+\right] \\ &= \sigma^2\text{tr}\left[\left(\bar{X}_i\right)^+ X_i X_i^T \left(\bar{X}_i\right)^{+T}\right] \\ &= \sigma^2\text{tr}\left[\left(\bar{X}_i\bar{X}_i^T\right)^+ X_i X_i^T\right] \\ &= \sigma^2\text{tr}\left[X_i^T \left(\bar{X}_i\bar{X}_i^T\right)^+ X_i\right] \end{aligned}$$

For $\bar{X}_i$, we have:

$$
\begin{aligned}
\bar{X}_i &= X_i + E_i \\
&= Z_i \Lambda_i^{1/2} U_i^T + \sigma W_i \\
&= \left( Z_i \Lambda_i^{1/2} + \sigma W_i U_i \right) U_i^T
\end{aligned}
$$

The elements of $Z_i$ and $W_i$ are sampled independently from standard Gaussian distributions. The last equality holds because standard Gaussian random vectors are invariant under orthonormal transformations. Then $Z_i \Lambda_i^{1/2} + \sigma W_i U_i$ is a Gaussian random matrix whose $(r, s)$-th entry is a Gaussian random variable with mean 0 and variance $\lambda_{ir}^2 + \sigma^2$.

Based on this observation, the variance term becomes:

$$
\text{Var}(\hat{\beta}_i) = \sigma^2 \text{tr} \left[ \sum_{r=1}^{d/k} (\lambda_{ir}^2 + \sigma^2) z_r^T \left( \bar{X}_i \bar{X}_i^T \right)^+ z_r \right]
$$

$$
\leq \sigma^2 \lambda_{\min}^{-1} \left( \bar{X}_i \bar{X}_i^T \right) \sum_{r=1}^{d/k} (\lambda_{ir}^2 + \sigma^2) z_r^T z_r
$$

We can bound $\lambda_{\min}^{-1} \left( \bar{X}_i \bar{X}_i^T \right)$ using the following lemma, since $\bar{X}_i \bar{X}_i^T = \sum_{r=1}^{d/k} (\lambda_{ir}^2 + \sigma^2) z_r z_r^T$:

[Lemma 5 in Li et al. (2020)] Let

$$
\hat{\mathbf{A}} = \sum_{i=1}^n \hat{\lambda}_i \mathbf{w}_i \mathbf{w}_i^T,
$$

where $\mathbf{w}_i \in \mathbb{R}^d$ is a random vector with each entry independently distributed as $\mathcal{N}(0, 1)$. There exists a universal constant $b_1$ such that with probability at least $1 - 2e^{-t}$, we have:

$$
\sum_{i=1}^n \hat{\lambda}_i - \Lambda \leq \mu_n(\hat{A}) \leq \mu_1(\hat{A}) \leq \sum_{i=1}^n \hat{\lambda}_i + \Lambda
$$

where

$$
\Lambda = b_1 \left( \hat{\lambda}_1 (t + n \log 9) + \sqrt{(t + n \log 9) \sum_{i=1}^n \hat{\lambda}_i^2} \right)
$$

Furthermore, there exists a universal constant $b_2$ such that with probability at least $1 - 2e^{-n/b_2}$:

$$
\frac{1}{b_2} \sum_{i=1}^n \hat{\lambda}_i - b_2 \hat{\lambda}_1 n \leq \lambda_{\min}(\hat{A}) \leq \lambda_{\max}(\hat{A}) \leq b_2 \sum_{i=1}^n \hat{\lambda}_i + b_2 \hat{\lambda}_1 n
$$

Assume there exists a $t^\star$ satisfying:

$$
t^\star = \min \left\{ 0 \leq j \leq n/k : \frac{\sum_{r=j+1}^{n/k} (\lambda_{ir}^2 + \sigma^2)}{\lambda_{i(j+1)}^2 + \sigma^2} > bn \right\}
$$

for some constant $b$. Then we have:

$$
\lambda_{\min} \left( \sum_{r=1}^{d/k} (\lambda_{ir}^2 + \sigma^2) z_r z_r^T \right) \geq \lambda_{\min} \left( \sum_{r=t^\star+1}^{d/k} (\lambda_{ir}^2 + \sigma^2) z_r z_r^T \right) \geq c_1 \sum_{r=t^\star+1}^{n/k} (\lambda_{ir}^2 + \sigma^2)
$$

for some constant $c_1$, due to Lemma A.6.2.

The term $\sum_{r=1}^{d/k} (\lambda_{ir}^2 + \sigma^2) z_r^T z_r$ can also be upper bounded since $z_r^T z_r$ concentrates around $n/k$:

$$
\sum_{r=1}^{d/k} (\lambda_{ir}^2 + \sigma^2) z_r^T z_r \leq c_2 n \sum_{r=1}^{d/k} (\lambda_{ir}^2 + \sigma^2)/k
$$

for some constant $c_2$ with high probability.

Therefore, we obtain an upper bound for the variance term with high probability:

$$\text{Var}(\hat{\beta}_i) \leq \sigma^2 c_3 \frac{n \sum_{r=1}^{d/k}(\lambda_{ir}^2 + \sigma^2)}{k \sum_{r=t^\star+1}^{n/k}(\lambda_{ir}^2 + \sigma^2)}$$

However, this upper bound is too loose as it can diverge as $n \to \infty$, while in practice the variance does not exhibit this behavior.

For the dense estimator, obtaining even such a crude bound is challenging. If we use the notation:

$$\bar{X}\bar{X}^T = \sum_{i=1}^{d} s_i s_i^T$$

where $s_r = \left[\sigma z_{r,1}^T \ldots \sqrt{(\lambda_{ir} + \sigma^2)} z_{r,i}^T \ldots \sigma z_{r,k}^T\right]$, then applying Lemma A.6.2 becomes impossible because we cannot extract a common scalar factor and construct columns consisting of independent and identically distributed standard Gaussian random variables.

### A.6.3  A CASE STUDY

In this section, we analyze a simplified case for sparse estimators where the number of features $d$ is equal to the number of experts $k$. This implies that each expert focuses on a single, unique dimension. We consider a one-dimensional scenario for this analysis. Assume the true feature $x \sim \mathcal{N}(0, \lambda^2)$, the observation error $e \sim \mathcal{N}(0, \sigma^2)$, and the response $y = x\beta$. The feature $x$ is observed with noise, so the regressor used is $x' = x + e$. We analyze the underparameterized case, ensuring at least one data sample per expert for estimation (here, $n \geq 1$ for the single parameter $\beta$).

The Ordinary Least Squares (OLS) estimator $\hat{\beta}$ for regressing $y_i$ on $x_i + e_i$ is:

$$\hat{\beta} = \frac{\sum_{i=1}^{n}(x_i + e_i)(\beta x_i)}{\sum_{i=1}^{n}(x_i + e_i)^2} = \beta \frac{\frac{1}{n}\sum_{i=1}^{n}(x_i^2 + x_i e_i)}{\frac{1}{n}\sum_{i=1}^{n}(x_i^2 + 2x_i e_i + e_i^2)}$$

The generalization error is:

$$\mathcal{R} = \mathbb{E}_{x_i, e_i}\left[\lambda^2(\hat{\beta} - \beta)^2 + \sigma^2 \hat{\beta}^2\right]$$
$$= \lambda^2 \mathbb{E}[(\hat{\beta} - \beta)^2] + \sigma^2 \mathbb{E}[\hat{\beta}^2]$$

Based on the law of large numbers, the bias term $\lambda^2 \mathbb{E}[(\hat{\beta} - \beta)^2]$ can be approximated by $\frac{\sigma^2 \lambda^2 \beta^2}{\lambda^2 + \sigma^2}$, which is exactly the Bayes error in this case. Then by the Delta method, the variance is:

$$\sigma^2 \mathbb{E}[\hat{\beta}^2] = \beta^2 \left(\frac{\lambda^2}{\lambda^2 + \sigma^2}\right)^2 \frac{1}{n}\left[\frac{2\lambda^4 + \lambda^2\sigma^2}{(\lambda^2)^2} + \frac{2(\lambda^2 + \sigma^2)^2}{(\lambda^2 + \sigma^2)^2} - \frac{2 \cdot 2\lambda^2(\lambda^2 + \sigma^2)}{\lambda^2(\lambda^2 + \sigma^2)}\right]$$
$$= \sigma^2 \beta^2 \left(\frac{\lambda^2}{\lambda^2 + \sigma^2}\right)^2 \frac{1}{n}\left[\frac{2\lambda^2 + \sigma^2}{\lambda^2} + 2 - 4\right]$$
$$= \sigma^2 \beta^2 \left(\frac{\lambda^2}{\lambda^2 + \sigma^2}\right)^2 \frac{1}{n}\left[\frac{2\lambda^2 + \sigma^2 - 2\lambda^2}{\lambda^2}\right]$$
$$= \sigma^2 \beta^2 \left(\frac{\lambda^2}{\lambda^2 + \sigma^2}\right)^2 \frac{1}{n}\frac{\sigma^2}{\lambda^2}$$
$$= \frac{\beta^2 \lambda^2 \sigma^4}{n(\lambda^2 + \sigma^2)^2}$$

However, our experiments (see Figure 6 and Table 1) indicate different convergence behaviors. For the dense estimator (not analyzed here), the excess risk fits well to an $O(n^{-2})$ curve. For the sparse estimator (analyzed in this section), the empirical fit includes both $O(n^{-1})$ and $O(n^{-2})$ terms. These empirical findings suggest that higher-order terms or other phenomena not captured

by this analysis play a significant role, underscoring the challenges in obtaining a precise theoretical analysis of sample complexity that fully matches experimental results.

In Table 1, we list the closed-form fitting curves for the excess risks of both dense and sparse estimators, corresponding to the experimental conditions shown in Figure 6.

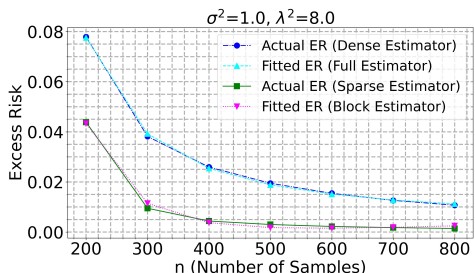
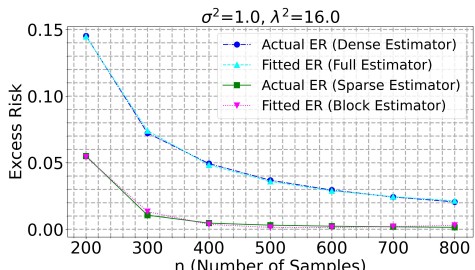
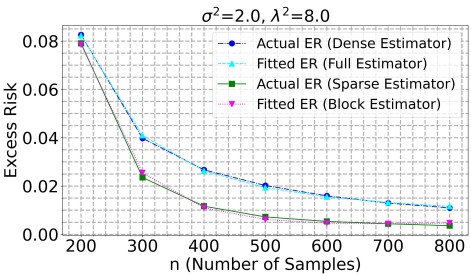
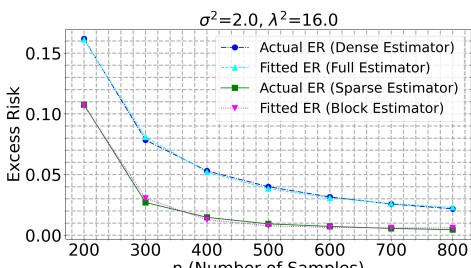

Figure 6: The actual and fitted curves for the excess risks of the dense estimator and sparse estimator under the setting $d = k = 100$.

Table 1: Closed-form fitting curves for the excess risks under the parameter setting $\sigma^2 = 1$ and $\lambda^2 = 8$. Four different runs are presented.

| Run | Dense estimator | Sparse estimator | |
|---|---|---|---|
| 1 | $\dfrac{2.67 \times 10^3}{n^2}$ | $\dfrac{3.94 \times 10^3}{n^2}$ | $- \dfrac{13.76}{n}$ |
| 2 | $\dfrac{4.86 \times 10^3}{n^2}$ | $\dfrac{5.24 \times 10^3}{n^2}$ | $- \dfrac{19.10}{n}$ |
| 3 | $\dfrac{2.92 \times 10^3}{n^2}$ | $\dfrac{5.51 \times 10^3}{n^2}$ | $- \dfrac{11.10}{n}$ |
| 4 | $\dfrac{5.64 \times 10^3}{n^2}$ | $\dfrac{11.7 \times 10^3}{n^2}$ | $- \dfrac{70.85}{n}$ |

## A.7 PROOF OF NEARLY PERFECT ROUTER

### A.7.1 FORMAL STATEMENT OF THEOREM 1

Consider $n$ samples arranged in matrix $\bar{X} \in \mathbb{R}^{n \times d}$ as in the paper. Assign labels $Y \in \{1, \ldots, k\}^n$ indicating block membership for each row. Given training data $(\bar{X}, Y)$, we train a classifier for test points $x_{\text{test}} \in \mathbb{R}^d$, as the router. We design a quadratic discriminant analysis (QDA) based classifier for it. To train such a QDA classifier, we assume we have access to $\sigma^2$ (estimation of $\sigma^2$ is easy) and denote the feature sets for each block as $\{S_i\}_{i=1}^k$. We assume a balanced loading, i.e., $n_i \approx n_j, \forall i, j$, for simplicity.

To estimate the covariance matrix for each block $i$, we need two steps: first, extract submatrix $\bar{X}_{i,S_i} \in \mathbb{R}^{n_i \times d_i}$ (rows with $Y_j = i$, columns $S_i$); second, compute sample covariance:

$$\hat{C}_i = \frac{1}{n_i}(\bar{X}_{i,S_i})^\top \bar{X}_{i,S_i}$$

For a test point $x_{\text{test}} \in \mathbb{R}^d$, we first need to compute the discriminant score for each class $i$:

$$\hat{g}_i(x_{\text{test}}) = -\frac{1}{2}\log|\hat{C}_i| - \frac{1}{2}x_{\text{test},S_i}^\top \hat{C}_i^{-1} x_{\text{test},S_i}$$

Then the predicting label should be $\hat{y} = \arg\max_{i \in \{1,\dots,k\}} \hat{\delta}_i(x_{\text{test}})$. Next is the sample complexity: for any $\epsilon > 0$, $\delta \in (0,1)$, if the total samples satisfy:

$$n \geq C\frac{M^2}{\epsilon^2}\sum_{i=1}^{k}\max\left\{d_i, \log\left(\frac{k}{\delta}\right)\right\}$$

then with probability at least $1 - \delta$, the excess risk $R - R^* \leq \epsilon$. Here $M = \max_i \|C_i\|_2 \leq \max_i \|\Sigma_i\|_2 + \sigma^2 < \infty$.

### A.7.2   PROOF

The excess risk can be upper bounded by Fan et al. (2012):

$$R - R^* \leq K\max_i \|\hat{C}_i - C_i\|_2$$

where $K > 0$ is a constant depending on the separation degree between classes. The estimation error of the sample covariance matrix can be upper bounded by Wainwright (2019):

$$\mathbb{P}\left(\|\hat{C}_i - C_i\|_2 \geq c_1 M\left(\sqrt{\frac{d_i}{n_i}} + \frac{d_i}{n_i}\right) + t\right) \leq c_2 \exp\left(-c_3 n_i \min\left\{\frac{t^2}{M^2}, \frac{t}{M}\right\}\right)$$

with $n_i$ samples, where $c_1, c_2, c_3 > 0$ are absolute constants. Denote $\eta = \epsilon/K$. If $n_i \geq C_1\frac{M^2 d_i}{\eta^2}$, where $C_1$ is a large enough constant compared to $c_1$, we have $c_1 M\left(\sqrt{\frac{d_i}{n_i}} + \frac{d_i}{n_i}\right) \leq \eta/2$. If we also have $n_i \geq \frac{4M^2}{c_3\eta^2}\log(\frac{c_2 k}{\delta})$, then the following inequality holds:

$$\mathbb{P}\left(\|\hat{C}_i - C_i\|_2 \geq \eta\right) \leq \frac{\delta}{k}$$

Then by the union bound, we can get the sample complexity result. This sample complexity result means that we can get a "perfect" router with large enough dataset.

## B   EXPERIMENTAL DETAILS

We provide details and hyperparameters used for the experiments.

### B.1   MODULAR STRUCTURE

The following methodology is used to generate the modular structure visualizations presented in the paper. We begin by applying the TEAL algorithm (Liu et al., 2024b) to obtain sparse activations. TEAL achieves substantial activation sparsity with minimal loss to model performance by identifying layer-specific thresholds and pruning input activations with magnitudes below those thresholds. Two thresholding strategies are available to achieve a target sparsity: (1) **uniform thresholding**, where the sparsity constraint is applied uniformly across the input activations of all linear layers in the model, and (2) **greedy thresholding**, where thresholds are selected per layer to minimize performance loss while meeting an overall sparsity constraint at the decoder block level. In our experiments, we use the precomputed thresholds provided in the TEAL GitHub repository (https://github.com/FasterDecoding/TEAL). All subsequent analysis is performed on the activations pruned using one of these two approaches.

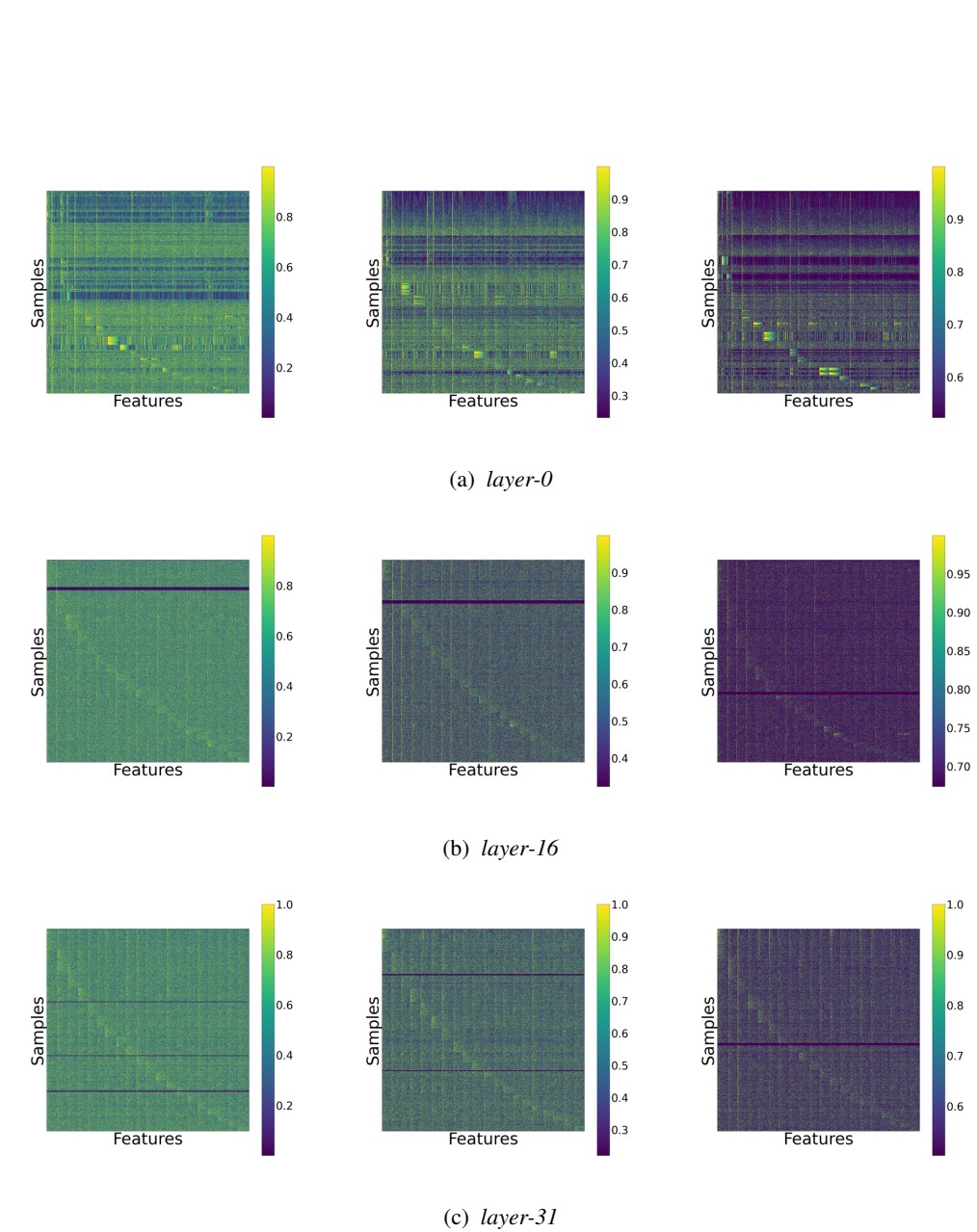

(a) *layer-0*

(b) *layer-16*

(c) *layer-31*

Figure 7: Modular structure in input activations to the up_proj layer within the MLP block of (a) *layer-0*, (b) *layer-16*, and (c) *layer-31* of the Llama-2-7B model, revealed using TEAL for greedy magnitude pruning. From left to right, panels show activation percentiles after pruning activations based on 0%, 40%, and 70% *greedy* activation sparsity thresholds, respectively.

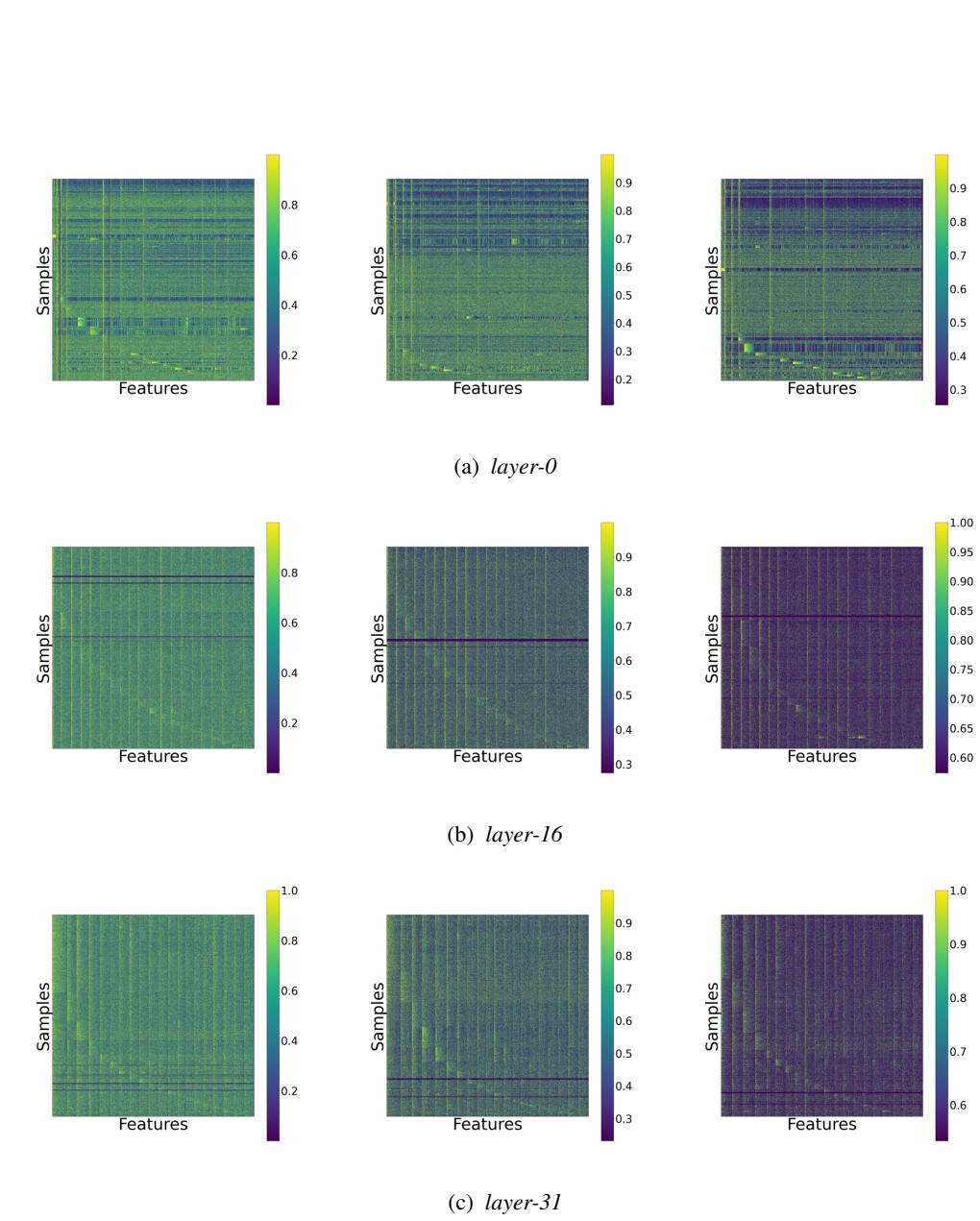

(a) *layer-0*

(b) *layer-16*

(c) *layer-31*

Figure 8: Modular structure in input activations to the up_proj layer within the MLP block of (a) *layer-0*, (b) *layer-16*, and (c) *layer-31* of the Llama-3.1-8B model, revealed using TEAL for greedy magnitude pruning. From left to right, panels show activation percentiles after pruning activations based on 0%, 40%, and 70% *greedy* activation sparsity thresholds, respectively.

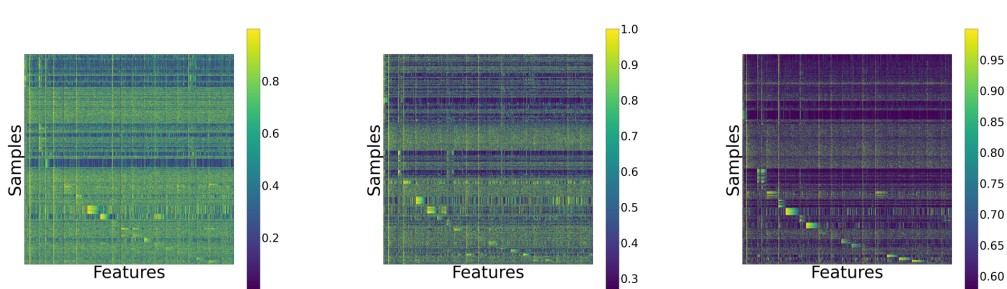

Figure 9: Modular structure in input activations to the gate_proj layer within the MLP block of *layer-0* of the Llama-2-7B model, revealed using TEAL for greedy magnitude pruning. From left to right, panels show activation percentiles after pruning activations based on 0%, 40%, and 70% *greedy* activation sparsity thresholds, respectively.

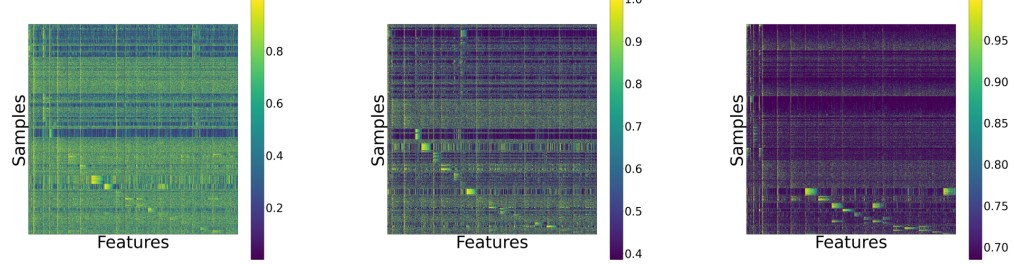

Figure 10: Modular structure in input activations to the up_proj layer within the MLP block of *layer-0* of the Llama-2-7B model, revealed using TEAL for uniform magnitude pruning. From left to right, panels show activation percentiles after pruning activations based on 0%, 40%, and 70% *uniform* activation sparsity thresholds, respectively.

To uncover the modular structure, we start by identifying feature blocks. We compute an affinity matrix over feature dimensions using cosine similarity and apply spectral clustering to group similar features into blocks. We divide features into 20 such modular for all experiments. Tokens are then assigned to the most compatible feature modular based on activation magnitudes, and both features and tokens are reordered such that the $i^{th}$ feature modular and its associated tokens form the $i^{th}$ diagonal block of the resulting feature matrix. Token-to-modular assignment is determined by ranking activation magnitudes and assigning each token to the modular where it obtains the highest average percentile rank across the block's features. For visualization, we use these percentile ranks rather than raw activation values to reduce the visual impact of extreme outliers, which can otherwise dominate the heatmap due to their magnitude being orders above the mean.

It is important to note that feature and token orderings are recomputed independently for each configuration (i.e., layer and sparsity level). As a result, the block diagonal structures seen in different visualizations are not directly comparable: each reflects a distinct clustering and reordering based on its specific activation pattern.

Figure 1 provides initial evidence for the emergence of modular structure in the input activations to the MLP block of layer 2 in the Llama-2-7B model. In all visualizations, activations are collected by passing tokens from the validation set of the Wikitext2 (`wikitext-2-raw-v1`) dataset through the model with a fixed sequence length of 1024. This design choice is consistent across all experiments. To assess the generality of the observed modular structure, we include further visualizations in Figures 7 and 8. These show activation patterns from the `up_proj` layers of the MLP blocks at the $0^{th}$, $16^{th}$, and $31^{st}$ decoder layers of both Llama-2-7B (Touvron et al., 2023) and Llama-3.1-8B models (Grattafiori et al., 2024), using greedy thresholding at 0%, 40%, and 70% sparsity levels. Across these settings, approximate modular-diagonal patterns persist. The structure is more pronounced in earlier decoder layers and at higher sparsity levels.

We also observe that this structure holds across both thresholding strategies. Figure 10 shows that modular structure is preserved even when using uniform thresholding. Additionally, Figure 9 illustrates similar patterns in the input activations to the `gate_proj` layers, further supporting the consistency of this phenomenon.

Each experiment requires two A100 GPUs (40GB memory) to accommodate model loading and activation storage.

## B.2 ROBUSTNESS TO NOISE

To generate the sparsely activated T5-base model using MoEfication, we use k-means clustering approach over feature dimensions and divide them into 96 experts. We choose activation sparsity levels corresponding to choosing 20,40,60, and 80 experts out of the 96. The validation set of SST2 dataset (Socher et al., 2013) is used for performance evaluation similar to (Zhang et al., 2021). We use RandomWordAug (word swap) and KeyboardAug augmentations from the NLPAug library (https://github.com/makcedward/nlpaug) to simulate word and character noise. As already mentioned, we perform at least one and a maximum of two swaps per sentence to simulate word noise. To simulate character noise, one character within each of two randomly chosen words is replaced with a keyboard error. We use a single A100 (40GB) GPU to get the activation sparse model and for further robustness evaluation.

Figure 11 shows both the accuracy under noisy and clean conditions, as well as the corresponding performance gap, with confidence intervals computed over 100 random seeds (0–99). The results provide empirical support for our hypothesis that sparsely activated models are more robust to noise than dense models. At intermediate activation sparsity levels (40 experts), not only is the performance gap lower than dense models, but the sparse models also outperform the dense models in the presence of noise.

## B.3 EXPERIMENTAL JUSTIFICATION OF THE LINEAR MODEL ASSUMPTION

To validate our theoretical insights in a more complex setting, we performed experiments on a synthetic dataset designed with a perfect modular structure. We trained both dense and MoE models on two tasks: a regression task with a single-layer linear network and a classification task with a

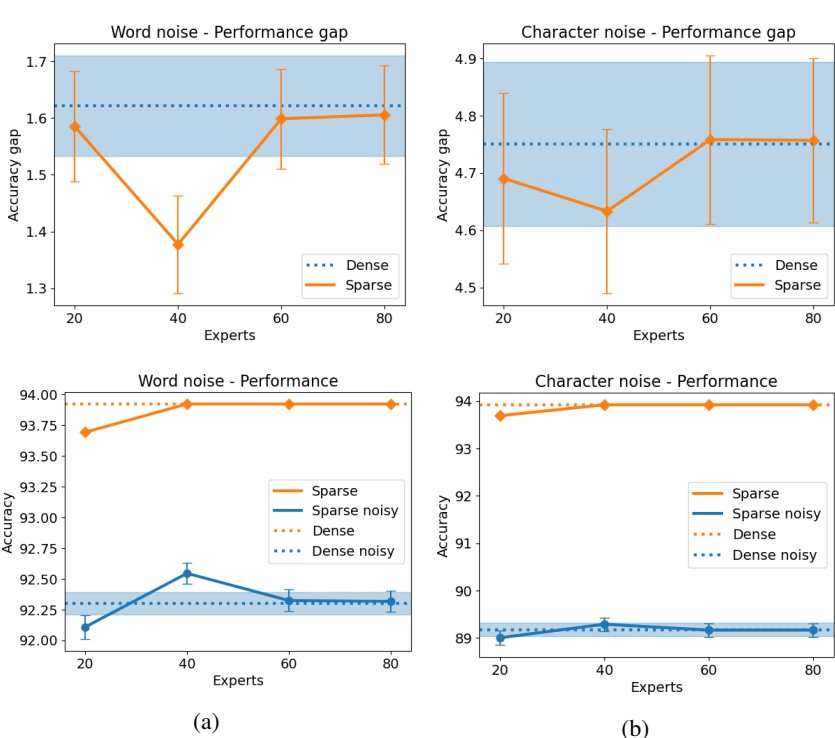

(a)  (b)

Figure 11: Robustness to noise. The figure shows the mean performance and confidence intervals of dense and activation sparse T5-base models and the accuracy gap between them as a result of noise over SST2 dataset over 100 seeds. (a) Word noise, (b) Character noise, (top) accuracy gap of dense and sparse models over clean and noisy datasets (bottom) accuracy. The confidence intervals for sparse models over noisy dataset are shown using error bars. The same is shown for dense models using the blue color fill.

two-layer non-linear network using ReLU activation. Both MoE models had access to a correctly routed input. We evaluated robustness by adding Gaussian noise $\mathcal{N}(0, \sigma^2)$ to the input features.

The results, shown in Table 2 and Table 3, demonstrate that MoE models are consistently more robust to noise than their dense counterparts across both linear and non-linear tasks. This supports our central claim that the robustness benefit is a fundamental property of the modular architecture.

Table 2: Robustness to noise (MSE). Lower MSE is better. The MoE model is more robust to noise, showing lower MSE and a smaller drop in performance.

| Noise Std ($\sigma$) | Dense w/ noise (MSE ↓) | MoE w/ noise (MSE ↓) | Dense-Robustness drop (↓) | MoE-Robustness drop (↓) |
|---|---|---|---|---|
| 0.1 | 3.13e-3 ± 4.45e-6 | 7.07e-4 ± 6.36e-6 | 2.10e-4 ± 2.18e-7 | -1.54e-5 ± 4.28e-6 |
| 0.2 | 9.88e-3 ± 6.85e-6 | 2.64e-3 ± 1.73e-5 | 3.41e-3 ± 3.45e-6 | 1.50e-5 ± 1.69e-5 |
| 0.3 | 1.57e-2 ± 1.17e-5 | 5.64e-3 ± 3.49e-5 | 1.41e-2 ± 1.49e-5 | 2.84e-4 ± 4.13e-5 |
| 0.4 | 1.88e-2 ± 2.44e-5 | 9.40e-3 ± 5.79e-5 | 3.39e-2 ± 2.47e-5 | 1.01e-3 ± 8.65e-5 |
| 0.5 | 2.03e-2 ± 5.37e-5 | 1.36e-2 ± 8.46e-5 | 6.15e-2 ± 4.07e-5 | 2.42e-3 ± 1.63e-4 |

Table 3: Robustness to noise (Accuracy) on a **2-layer non-linear classification task** ($\sigma = 0.1$). Higher accuracy is better. The MoE model achieves higher accuracy, and its performance drops less than the dense model's.

| Seed | Dense w/ noise (Acc. ↑) | MoE w/ noise (Acc. ↑) | Dense-Robustness drop (↓) | MoE-Robustness drop (↓) |
|---|---|---|---|---|
| 1 | 74.49% | 76.96% | 1.03% | 0.16% |
| 2 | 78.06% | 79.42% | 0.68% | 0.15% |
| 3 | 74.68% | 75.96% | 0.01% | -0.00% |
| 4 | 73.02% | 74.60% | 0.35% | 0.08% |

