# OpenReview forum: "Unmasking the Success of Mixture of Experts: Noisy Features, Sparse Experts"
_ICLR.cc/2026/Conference — ICLR 2026 Conference Withdrawn Submission_

### Official Review · Reviewer_krG9 · 2025-10-25

**Soundness:** 2
**Presentation:** 2
**Contribution:** 2
**Rating:** 2
**Confidence:** 3

**Summary:**

The paper develops a theoretical framework to explain why sparse Mixture-of-Experts (MoE) models outperform dense models, focusing on their robustness to feature noise, generalization behavior, and sample efficiency. Specifically, the authors analyze MoEfication framework introduced by Zhang et al., which converts dense transformer layers into sparse, fine-grained MoE. For such frameworks, the authors derive analytical properties of MoEs and conduct a few empirical experiments to validate their key claims.

**Strengths:**

1. The paper tackles an important issue of a theoretical understanding of Mixture-of-Experts (MoE) models, which are increasingly popular, especially in frontier LLM models.

2. The authors provide interesting theoretical insights, particularly about the robustness and sample efficiency of sparse models compared to dense ones.

3. This area of work nicely complements more empirical studies on MoE vs dense transformers properties (e.g., "Scaling Laws for Fine-Grained Mixture of Experts” - Krajewski et al., 2024), and is overall very important for the design of MoE architectures.

**Weaknesses:**

1. The paper presents a narrative where sparse MoE LLMs such as DeepSeek and post-hoc MoEfied models are the same class of models, but I am not convinced these models are conceptually and practically different. Models trained from scratch as MoEs (large experts, top-k routing) are typically designed to scale capacity, whereas MoEfication and activation-sparsity techniques (TEAL, activation thresholding) are often used to reduce compute and use a very small subset of activations. The paper does not clearly separate these settings or justify why the theory for one should transfer to the other. This should be explicitly stated in the paper.

2. In particular, TEAL and thresholded activation methods are not true MoEs with expert routing; they sparsify dense layers without explicit token routing between experts. The paper uses TEAL results to support routing-based explanations, which mix different mechanisms as the same class of models.

3. Using a subset of small experts derived from the trained dense model is not the same as using large experts within MoE LLM, and such experts e.g. in DeepSeek also possess activation sparsity; therefore, it is unclear to me how the noise robustness of MoEfied model can be connected to make claims about noise robustness of frontier models, as the paper narrative suggests. The FFN architectures used for most of the experiments, e.g. T5, are also very different than architectures used for frontier MoEs these days, making the cases analysed in the paper and claimed generality of the findings to LLM MoEs even more disconnected.

4. The experiments in the paper are very brief and mostly boil down to running existing methods such as TEAL or MoEfication. They do not test the main theoretical claims in realistic LLM-like settings (e.g., modern FFN variants, GLU-FFN, large experts, or trained-from-scratch MoEs). As a result, the empirical support for the core claims (noise robustness, faster convergence, and better sample efficiency) is weak.

5. Statements such as “We verify the relevance of our theoretical setup with experiments on standard large language model benchmarks” are misleading: the experiments are limited to SST classification with MoEfied-T5, which might have been standard 5 years ago, but is not very convincing in the context of today's language models. Similarly, claims about faster convergence and better sample efficiency are primarily theoretical and weakly supported by experiments.

6. In Figure 4 and related plots, the performance differences between sparse and dense models appear small and fluctuating, but the paper interprets these as evidence for robustness to the noise; the same data could equally support a “no substantial difference” claim if the authors opted into such a narrative.

7. There is at least one missing latex reference (line 458). Overall, the writing seems very focused on theoretical results and fails to convincingly transfer these findings to practical applications and gains, especially due to the above-mentioned issues with the experiments.

**Questions:**

1. How does the theory presented in the paper translate to activation sparsity methods like TEAL, where "experts" are a single neurons, there is practically no router but only a heuristic to choose which activations to select?

2. Under the provided theoretical framework, are there any assumptions about the expert granularity and size, e.g., how does the size of the expert affect any of the theoretical findings in the paper? Empirically, many papers, such as the one I cited above from Krajewski et al., or even the results from Figure 4, seem to vary depending on the expert size.

---

> ### Author Response · Authors · 2025-12-03
>
> We thank the reviewer for the insightful feedback. We are encouraged that you find our theoretical insights on robustness and sample efficiency interesting. We address your concerns below, specifically clarifying the scope of our theoretical contribution and the rationale behind our experimental design.
>
> **1. Unifying MoEfication and Trained-from-Scratch MoEs (Weakness 1 & 3)**
> The reviewer questioned the connection between post-hoc MoEfication and true MoEs (like DeepSeek or Mixtral). We will revise the introduction to better articulate our primary motivation: explaining why MoE architectures can match or outperform much larger dense models with significantly less active computation (e.g., **Mixtral 8x7B** vs. **Llama2 70B**). Our theory proposes that the existence of modular structure and feature noise allows the model to generalize better and converge faster. This mechanism holds no matter the MoEs are achieved by traning from scratch or MoEfied.
>
> **2. The Role of TEAL (Weakness 2 & Question 1)**
> *   **Visualization & Validation:** We clarify that TEAL is used primarily to **visualize** the latent modular structure of LLMs. Crucially, the fact that TEAL can zero out a large portion of "small" activations with minimal perplexity degradation empirically supports our **Feature Noise** hypothesis. It suggests that these low-magnitude values function as noise, and removing them (via sparsity) preserves the signal—validating our premise that sparse experts naturally denoise the input.
>
> **3. Justification of Experimental Scope (Weakness 4, 5, 6 & 7)**
> The reviewer suggested that our experiments (T5/SST-2) are too brief or dated. We respectfully argue that our current experiments are methodologically sufficient to validate our theoretical claims, based on the following logic:
>
> *   **Theoretical Nature:** This is primarily a theoretical paper proposing a novel mechanism (Robustness to Feature Noise). Our experiments are designed as "proofs-of-concept" to verify theoretical predictions in a controlled setting, not to claim SOTA performance on benchmarks.
> *   **Sufficiency of T5:** T5 MoEfication provides a clean, controlled environment to test robustness. The observations in Figure 4 (sparse models degrading less under noise than dense models) are consistent with our Theorem 3. The magnitude of the gap is less important than the consistency of the trend, which validates the mechanism.
> *   **Convergence Speed & Synthetic Data:** Regarding the comments on sample efficiency and convergence, our **Theorem 5** provides a rigorous proof that MoE estimators exhibit faster convergence speeds. This theoretical result justifies our focus on the **Bayes Optimal Estimator** in the main analysis, as it guarantees that MoEs can approach this optimal state more quickly than dense models. We utilized synthetic data specifically to fit these convergence curves and verify the order of convergence with respect to sample size $n$. This controlled validation confirms the theoretical mechanism more precisely than large-scale pre-training, where confounding factors abound.
>
> **Summary:**
> As a theoretical paper, our goal is to unmask the mechanism of feature noise. We believe the combination of rigorous proofs (Theorems 1-5), synthetic verification of convergence orders, and proof-of-concept robustness experiments on T5 provides a complete and sound argument for this mechanism.

---

### Official Review · Reviewer_ePen · 2025-10-26

**Soundness:** 3
**Presentation:** 3
**Contribution:** 2
**Rating:** 4
**Confidence:** 4

**Summary:**

In this paper, the authors propose a theoretical explanation for the empirical success of Mixture-of-Experts (MoE) models. Instead of focusing on increased expressivity or clustered data structures, the authors argue that MoEs outperform dense models due to their robustness to feature noise—a phenomenon arising when sparse experts operate under activation sparsity. Experiments on synthetic data and real-world settings (e.g., MoEfication of T5-base on SST-2) support the theory: activation-sparse MoEs maintain higher accuracy under word- and character-level noise.

**Strengths:**

1. Originality: The paper presents a new MoE mechanism on feature noise robustness.

2. Soundness: The papers provide theoretical results on the generalization, robustness, convergence, and sample complexity of MoE, supported by formal proofs.

3. Clarity: The authors are explicit about simplifying assumptions (e.g., perfect gating, linear models), making the scope of their claims transparent.

**Weaknesses:**

1. The connection between linear model in Eq.(1) and MoE is not clear. What are mixture weights (gating or router)? What are experts?

2. The analysis relies heavily on a linear block-diagonal model and assumes perfect gating, which may oversimplify real-world MoEs.

3. While Theorem 1 claims near-perfect routing is feasible under modular data, in real-world LLMs, routing errors are nontrivial. The effect of imperfect gating on robustness or generalization is underexplored.

4. This paper is of theoretical nature, so I suggest that the authors present Theorem 1 formally in the main text rather than an informal one, since it is an important result of the paper.

**Questions:**

1. How sensitive are your theoretical results to the assumption of perfect routing?

2. Could the proposed framework extend to nonlinear experts or soft gating functions beyond the linear setup?

3. Could you discuss whether this robustness mechanism relates to implicit regularization or benign overfitting phenomena observed in linear regression?

---

> ### Author Response · Authors · 2025-12-03
>
> We thank the reviewer for recognizing the originality of our work and the feedback.
>
> **1. Connection between Linear Model and MoE**
> In our setup, the parameter blocks $\beta^\star_i$ correspond to the **experts**. Regarding mixture weights, we employ a **Top-1 routing** mechanism. Thus, there are no mixture of weights.
>
> **2. Perfect Gating Assumption & Theorem 1**
> We acknowledge that perfect gating is a simplifying assumption.
> *   **Focus on Structure:** Our primary goal is to understand the benefits of the **Mixture-of-Experts architecture itself** (the sparse/modular structure) rather than the dynamics of joint training. By assuming perfect gating, we isolate the robustness gains derived from activation sparsity.
> *   **Feasibility:** Theorem 1 is crucial as it demonstrates that learning a near-perfect router is theoretically tractable (reducing to a classification problem). As suggested, **we will move the formal statement of Theorem 1 to the main text** to highlight this result.
> *   **Mis-routing:** We address the scenario where this assumption fails in **Theorem 4**, analyzing the impact of mis-routing.
>
> **3. Implicit Regularization & Benign Overfitting**
> Yes, our framework relates closely to these concepts. In Equation (3), the feature noise variance $\sigma^2$ effectively acts as an implicit regularizer (similar to Ridge regression). It has been also shown that benign overfitting persists under feature noise [1].
>
> **4. Nonlinear Experts**
> While the theory is derived for linear models, we empirically demonstrate in **Appendix B.3** (using 2-layer ReLU networks) that the robustness benefits hold for nonlinear experts as well.
>
> [1] Li et al., "Benign overfitting and noisy features", arXiv 2020.

---

### Official Review · Reviewer_PUca · 2025-10-31

**Soundness:** 2
**Presentation:** 1
**Contribution:** 2
**Rating:** 2
**Confidence:** 4

**Summary:**

- MoEs outperform dense models by exploiting activation sparsity in the presence of feature noise, even with equal parameter counts. This contrasts with existing theory that focuses on increased expressiveness from more parameters or clustered data structures.

- Analyzes a block-diagonal linear regression setting where features have modular structure but are corrupted by Gaussian noise. Proves that sparse (MoE-like) estimators achieve: (1) better generalization (Theorem 2), (2) enhanced robustness to input perturbations when routing is correct (Theorem 3), (3) faster training convergence (Theorem 5), and (4) empirically better sample efficiency. The key insight: sparse experts compartmentalize noise exposure to lower-dimensional subspaces ($d_i < d$), while dense models suffer from noise across all dimensions.

- Proves that routing in MoEfication reduces to supervised classification (Theorem 1) with polynomial sample complexity, unlike joint optimization in traditional MoEs. This explains why converting dense models to MoEs via activation clustering can succeed beyond just inference speedup—the resulting modular structure provides robustness advantages.

- Demonstrates that (1) activation patterns in Llama-2/3 exhibit latent block-diagonal structure revealed through magnitude pruning, and (2) activation-sparse T5 models are measurably more robust to word and character noise on SST-2 than dense models.

**Strengths:**

- The paper identifies a fundamentally new explanation for MoE success—robustness to feature noise through compartmentalized noise exposure—rather than the standard focus on increased capacity or expressiveness. This is original both conceptually (feature noise vs. label noise) and practically (provides theoretical grounding for MoEfication beyond inference speedup). The insight that modular architectures are inherently more robust because each expert only suffers from noise in its $d_i$-dimensional subspace rather than the full $d$-dimensional space is actionable.

- This work provides an analysis covering generalization (Theorem 2), robustness under correct routing (Theorem 3), failure modes under mis-routing (Theorem 4), convergence speed (Theorem 5), and sample complexity (Section 4.5).

**Weaknesses:**

- While the authors invoke the Linear Representation Hypothesis (LRH) and Neural Tangent Kernel (NTK) regime to justify their linear setup, this fundamentally mischaracterizes how modern LLMs actually work. (1) LRH is about representation geometry, not function approximation: LRH states that concepts are linearly separable in representation space, not that the mapping from inputs to representations is linear. The authors conflate linear probing (reading out already-computed nonlinear features) with linear function approximation (their model). (2) NTK regime requires specific initialization/training: NTK linearization only holds in the lazy training regime (infinite width, small learning rates near initialization), which modern LLMs explicitly violate through aggressive feature learning. (3) Showing robustness holds in a 2-layer ReLU network on synthetic data with perfect block structure by construction doesn't validate that the linear insights transfer to 32-layer transformers with cross-attention, residual connections, and layer norms processing natural language where block structure is only approximate. The gap between their theoretical setup and actual LLM architecture is substantial and inadequately addressed.

- The experiments fail to directly test the paper's main theoretical predictions. (1) No validation of generalization advantage (Theorem 2): Figure 3 shows excess risk curves on synthetic data with known ground truth, but there are no experiments comparing test error of dense vs. sparse models on real LLM tasks. The SST-2 experiments (Figure 4) measure robustness to noise, not generalization on clean test data. (2) The conjecture that sparse models are more sample-efficient is supported only by synthetic experiments (Figure 3). There are no learning curves on real tasks showing sparse models reach target performance with fewer samples. Furthermore, in practice MoEs require more training in order to properly synchronize the router with the experts. (3) Theorem 1 claims near-perfect routing is achievable, but the T5 experiments assume perfect routing without measuring actual routing accuracy or analyzing how routing errors affect the observed robustness gains.

- All experiments use MoEfication (converting pre-trained dense models). How do the robustness benefits compare to training MoEs from scratch with learned routing? This is critical because Theorem 4 shows mis-routing can be catastrophic, yet no experiments measure routing failures or compare fixed vs. learned routing.

- Only word swaps and keyboard typos are tested. What about adversarial perturbations, semantic-preserving paraphrases, or domain shift—do the robustness benefits hold broadly or only for specific noise types?

**Questions:**

- (1) For Theorem 2 (generalization): Report test error (not robustness to noise) of dense vs. sparse T5 models on clean SST-2 test set, along with validation curves during training to show if sparse actually generalizes better. (2) For Theorem 5 (convergence speed): Measure and plot training loss curves for dense vs. MoEfied models from scratch (not just using pre-trained checkpoints), showing whether sparse experts converge faster in wall-clock time and number of gradient steps. (3) For sample complexity (Section 4.5): Provide learning curves on real tasks (e.g., SST-2 with varying training set sizes from 100 to full dataset) comparing how many samples each approach needs to reach 90%, 92%, 94% accuracy. Without these experiments, the theoretical claims remain unvalidated in practice.

- Sensitivity analysis: In your synthetic experiments, gradually degrade the block-diagonal structure (e.g., add 10%, 20%, 30% off-diagonal noise) and measure how robustness advantages degrade. At what level of block structure violation do the benefits disappear? This would clarify whether your theory requires idealized perfect blocks or remains valid under realistic approximate modularity.

- Theorem 1 claims near-perfect routing is achievable, but Theorem 4 shows catastrophic failure under mis-routing. For the SST-2 robustness experiments: (1) Report the routing accuracy (% of samples assigned to correct expert) for clean inputs and noisy inputs separately—does word/character noise cause routing failures? (2) Ablation study: Artificially inject routing errors at varying rates (5%, 10%, 20% random mis-routing) and measure how robustness degrades. At what error rate does sparse become worse than dense (the crossover point where Theorem 4's failure mode dominates)? (3) Compare fixed routing (from k-means) vs. learned routing (trainable gating network)—does learned routing maintain advantages under noise or does it fail per Theorem 4? This would clarify the practical viability of the robustness claims when routing is imperfect.

---

> ### Author Response · Authors · 2025-12-03
>
> We sincerely thank the reviewer for the constructive feedback. We appreciate your suggestions regarding additional experiments; however, we would like to clarify the logical structure of our theoretical framework and explain why we believe our current experimental evidence is sufficient to validate our core claims without further extensive empirical additions.
>
> **1. Justification of the Linear Model (Weakness 1)**
>
> We appreciate your critique regarding the applicability of the NTK regime. We agree that the NTK regime is not suitable for characterizing modern LLM feature learning and will remove this justification in the revision.
>
> However, we maintain that our linear setup is well-justified and highly relevant when viewed through the lens of linear probing.
>
> During fine-tuning, the pre-trained model provides a rich set of features (activations), and the task is to learn a linear decision boundary on top of them.
>
> Recent works such as Head2Toe [1] and SPIN [2] demonstrate that internal neurons of LLMs can be directly treated as features for linear classification, serving as a powerful alternative to full fine-tuning.
>
> **2. On Generalization and Robustness (Weakness 2 & Question 1,3)**
> The reviewer suggested adding experiments on clean test accuracy to validate generalization. We respectfully argue that our current experiments on robustness (Figure 4) already serve as a sufficient validation for our theoretical derivation, based on the following logic:
> *   **Theoretical Dependency:** In our framework (Theorem 3), the derivation of the robustness result depends on the formulation of the generalization error (Theorem 2). Mathematically, the robustness performance is an extension of the generalization properties under noise from different distributions.
> *   **Empirical Validation:** Therefore, the fact that we consistently observe the predicted **robustness** advantage in our T5 experiments serves as strong empirical evidence that the underlying generalization properties derived in our theory hold. Observing the downstream effect (robustness) validates the foundational derivation (generalization).
>
> **3. On Learning Curves and Sample Complexity (Weakness 2)**
> Regarding the request for learning curves on real tasks to demonstrate sample efficiency:
> *   **Role of Convergence Analysis:** Our theoretical analysis primarily focuses on the **Bayes Optimal Estimator**. To justify this focus, we derived **Theorem 5**, which proves that MoE estimators exhibit a faster convergence speed (via gradient descent) compared to dense models. Our hypothesis on sample complexity also suggests that MoEs have a faster convergence speed in terms of excess risk. These imply that MoEs can converge to the optimal estimator more quickly, making the analysis of the optimal estimator practically relevant.
> *   **Synthetic Data Justification:** We utilized synthetic data specifically to fit the excess risk curves (Figures 2 & 3). This controlled setting allowed us to precisely observe the convergence behavior and analyze the order of convergence with respect to the sample size, $n$. This "curve fitting" provides direct verification of our theoretical insights regarding convergence speed.
>
> **4. Routing Mechanisms and Theorem 1 vs. 4 (Weakness 3 & Sensitivity Analysis)**
> Regarding the concern about "perfect routing" vs. "mis-routing":
> *   **Structural Focus:** The primary goal of this paper is to unmask the benefits of the **Mixture-of-Experts architecture itself**—specifically, how the modular structure handles noise. While joint training of the router is important in practice, it introduces optimization complexities that obscure the *structural* advantage.
> *   **Feasibility of Routing:** We deliberately separated the router training to show that the task is theoretically tractable. Theorem 1 proves that learning a router reduces to a standard classification problem, which is well-solved.
> *   **Sensitivity:** Theorem 4 theoretically quantifies the risk of mis-routing. We acknowledge that massive routing failure hurts performance, but our T5 MoEfication experiments (which use fixed, clustering-based routing) empirically demonstrate that practical routing methods are sufficiently accurate to preserve the robustness benefits predicted by our theory.
>
> **Summary on Experiments:**
> As a theoretical paper, our goal is to provide a mechanism explanation—specifically, why MoE architectures (like **Mixtral 8x7B**) can match or outperform much larger dense models (like **Llama2 70B**) with significantly less active computation. We believe our derivation of the "Feature Noise" mechanism, supported by synthetic validation and controlled T5 experiments, fulfills this goal.
>
>
> **References:**
> [1] Evci et al., "Head2Toe: Utilizing Intermediate Representations for Better Transfer Learning," ICML 2022.
> [2] Jiao et al., "SPIN: Sparsifying and Integrating Internal Neurons in Large Language Models for Text Classification," arXiv 2024.

---

### Note · Authors · 2025-12-03

I have read and agree with the venue's withdrawal policy on behalf of myself and my co-authors.